health and disease and epidemiology/
mathematical modelling

COVID-19, SARS-n-COV, metapopulation epidemic models, network model, compartmental models, SEIAR model, nowcasting

**Author for correspondence:**
Rama Cont
e-mail: Rama.Cont@maths.ox.ac.uk

# Modelling COVID-19 contagion: risk assessment and targeted mitigation policies

Rama Cont, Artur Kotlicki and Renyuan Xu

Oxford University, Mathematical Institute, Oxford, UK

(iD) RC, 0000-0003-1164-6053; AK, 0000-0001-8707-2872;
RX, 0000-0003-4293-3450

We use a spatial epidemic model with demographic and geographical heterogeneity to study the regional dynamics of COVID-19 across 133 regions in England. Our model emphasizes the role of variability of regional outcomes and heterogeneity across age groups and geographical locations, and provides a framework for assessing the impact of policies targeted towards subpopulations or regions. We define a concept of efficiency for comparative analysis of epidemic control policies and show targeted mitigation policies based on local monitoring to be more efficient than country-level or non-targeted measures. In particular, our results emphasize the importance of shielding vulnerable subpopulations and show that targeted policies based on local monitoring can considerably lower fatality forecasts and, in many cases, prevent the emergence of second waves which may occur under centralized policies.

## 1. Overview

The novel coronavirus pandemic of 2019–2021 has led to disruption on a global scale, leading to more than 1.4 million deaths worldwide at the time of writing, and prompted the implementation of government policies involving a variety of 'non-pharmaceutical interventions' [1] including school closures, workplace restrictions, restrictions on social gatherings, social distancing and, in some cases, general lockdowns for extended periods. This has led to a range of different public health policies across the world, and the efficiency of specific policy choices has been subject to much debate.

While the nature of these restrictions has been justified by the severe threat to public health posed by the virus, their design and implementation necessarily involves a trade-off, often implicit in the decision-making process, between health outcomes and the socio-economic impact of such social restrictions.

An important feature of the COVID-19 pandemic has been the heterogeneity of epidemic dynamics and the resulting mortality across different regions, age classes and population categories. The importance of these heterogeneities suggests that homogeneous models—often invoked in discussions on reproduction number and herd immunity—may provide misleading insights, and points to the need for more granular modelling to take into account geographical, demographic and social factors which may influence epidemic dynamics.

We propose a flexible modelling framework which can serve as a decision aid to policy-makers and public health experts by quantifying this trade-off between health outcomes and social cost. Using a structured population model for epidemic dynamics which accounts for geographical and demographic heterogeneity, we formulate this trade-off as a control problem for a partially observed distributed system and provide a quantitative framework for comparative analysis of various mitigation policies. We illustrate the usefulness of the framework by applying it to the study of COVID-19 dynamics across regions in England and showing how it may be used to reconstruct the latent progression of the epidemic and perform a comparative analysis of various mitigation policies through scenario projections.

Several recent studies have used homogeneous compartmental models [2–9] or age-stratified versions of such models [10–15] to analyse the dynamics and impact of the COVID-19 epidemic in various countries. Our framework, while compatible with such homogeneous models at aggregate level, accounts for demographic and spatial heterogeneity in a more detailed manner, leading to regional outcomes which may substantially deviate from homogeneous models. Similar, though somewhat less detailed, heterogeneous models have been recently used to study COVID-19 outbreaks by Danon *et al.* [16] for the UK, Birge *et al.* [17] for New York City and Roques *et al.* [18] for France.

We first present below an overview of the main features of our approach and the key findings, before going into more detail on the methodology and results.

## 1.1. Methodology

We formulate a stochastic compartmental (SEIAR) epidemic model with spatial and demographic heterogeneity (age stratification) for modelling the dynamics of the COVID-19 epidemic and apply this model to the study of COVID-19 dynamics across regions in England.

The model takes into account:

— epidemiological features estimated by previous studies on COVID-19;
— the lack of direct observability of the total number of infectious cases and the presence of a non-negligible fraction of asymptomatic cases;
— the demographic structure of UK regions (age distribution, density);
— social contact rates across age groups derived from survey data;
— data on inter-regional mobility; and
— the presence of other random factors, not determined by the above.

We first demonstrate that this model is capable of accurately reproducing the early regional dynamics of the disease, both pre-lockdown and a month into lockdown, using a detailed calibration procedure that accounts for demographic heterogeneity across regions, low testing rates, and existence of asymptomatic carriers. The calibration reveals interesting regional patterns in social contact rates before and during lockdown.

Underlying any public health policy is a trade-off between a health outcome—which may relate to mortality or hospitalizations—and the socio-economic impact of measures taken to mitigate the magnitude of the impact on public health. We present an explicit formulation of this trade-off and use it to perform a comparative analysis of various 'social distancing' policies, based on two criteria:

— the benefit, in terms of reduction in projected mortality; and
— the cost, in terms of restrictions on social contacts.

The goal of our analysis is to make explicit the policy outcomes for decision-makers, without resorting to (questionable) concepts such as the 'economic value of human life' used in some actuarial and economic models [6,9,10].

In our comparative analysis, we consider a broad range of policies and pay particular attention to population-wide versus *targeted* mitigation policies, feedback control based on the number of observed cases. We introduce a concept of *efficient* policy, and show how this concept allows to identify

decision parameters which lead to the most efficient outcomes for each type of mitigation policy. The granular nature of our model, together with validation based on epidemiological data, provide a more detailed picture of the relative merits of various public health policies.

## 1.2. Summary of findings

Our first set of results concerns the reconstruction of the progression of the pandemic in England, in particular its latent spread through asymptomatic carriers.

— Using a baseline epidemic model consistent with epidemiological data and observations on fatalities and cases reported in England up to June 2020, we estimate more than 17.8 million persons in England (31.7% of the population) to have been exposed to COVID-19 by 1 August 2020. These estimates are much higher than numbers discussed in media reports, based on the number of *reported* cases.
— Based on a comparison of fatality counts and reported cases, we infer that less than 5% of cases in England had been detected prior to June 2020. This low detection probability implies in particular that the number of reported cases may severely underestimate the latent progression of the epidemic.
— We observe significant differences in epidemic dynamics across regions in England, with higher fatality and contagion levels in northern regions compared to southern regions, both before and during the lockdown period, pointing to the importance of demographic and geographical heterogeneity for modelling the impact of COVID-19.

After calibrating the model to replicate the regional progression of COVID-19 in England for the period 1 March to 31 May 2020, we use it for scenario projections under various mitigation policies. Comparative analysis of mitigation policies reveals that measures targeting subpopulations—such as regions with outbreaks—are more efficient than population-wide measures in terms of the trade-off between health outcomes and social cost. More specifically:

— Shielding of elderly populations is by far the most effective single measure for reducing the number of fatalities.[1]
— By contrast, school closures and workplace restrictions are seen to be less effective than social distancing measures outside of school and work environments.
— Adaptive policies (feedback control) which trigger measures when the number of daily observed cases exceed a threshold, are shown to be more effective than pre-planned policies, leading to a substantial improvement in health outcomes.
— A decentralized policy which triggers regional confinement measures based on regional daily reported cases is found to be more efficient than centralized policies based on national indicators, resulting on average in an overall reduction of 20 000 in fatalities and, in many cases, significant damping of a 'second wave'.
— Comparative analysis of policies (table 10) shows a wide range of health outcomes. The most effective policy in terms of reducing fatalities involves triggering of regional confinement measures based on monitoring of new cases, coupled with shielding of elderly populations.

Parameter uncertainty is an important issue in epidemic modelling. We perform robustness checks with respect to parameter uncertainty for various model parameters, most notably the symptomatic ratio and the infection rates, and show our policy comparisons to be robust with respect to various assumptions on these parameters.

The present work should be seen as an illustration of what may be done using our methodology, rather than an exhaustive analysis of different policy options and scenarios. We have made available an online implementation of the model, which may be used to explore other scenarios and policies than those presented below: http://covid19.kotlicki.pl.

## 1.3. Outline

The modelling framework is described in §2. Data sources and parameter estimations are detailed in §3. Section 4 highlights the implications of partial observability of state variables and the associated model uncertainty.

---

[1]We do recognize that the implementation of such shielding measures may be extremely challenging in practice.

The outcomes of various epidemic control policies are then discussed in §§5 and 6. Pre-planned policies are discussed in §§5.1 and 5.2, while §6 discusses *adaptive* (feedback) control policies, in which measures are triggered when the estimated number of new reported cases exceeds a threshold, and concludes with a comparative analysis of health outcomes and social cost of various types of mitigation policies.

# 2. Modelling framework

To take into account the role of geographical and demographic heterogeneity, we use a stochastic compartmental (SEIAR) model with age stratification, mobility across sites, social contact across age stratification, and the impact of asymptomatic infected individuals. For general concepts on deterministic and stochastic compartmental models, we refer to Anderson & May [19], Brauer & Castillo-Chavez [20], Britton *et al.* [21], Lloyd & Jansen [22].

## 2.1. State variables

We consider a regional metapopulation model with $K$ regions labelled $r = 1, \ldots, K$. Each region $r$ has a population $N(r)$ which is further subdivided into $M$ age classes labelled $a \in \{1, 2, 3, 4, \ldots, M\}$. We denote $N(r, a)$ the population in region $r$ in age category $a$, with $\sum_{a=1}^{M} N(r, a) = N(r)$.

Individuals in each region and age group are categorized into six compartments:

— Susceptible ($S$) individuals who have not yet been exposed to the virus.
— Exposed ($E$) individuals who have *contracted* the virus but are not yet infectious. Exposed individuals may then become *infectious* after a certain incubation period.
— Infectious ($I$) individuals who manifest symptoms.
— Asymptomatic ($A$) infectious individuals.
— Recovered ($R$) individuals. In line with current experimental and clinical observations on COVID-19, we shall assume that individuals who have recovered have temporary immunity, at least for the horizon of the scenarios considered, and cannot be re-infected [23].
— Deceased ($D$) individuals.

The progression of the disease in the population is monitored by keeping track of the respective number

$$S_t(r, a), \quad E_t(r, a), \quad I_t(r, a), \quad A_t(r, a), \quad R_t(r, a) \quad \text{and} \quad D_t(r, a)$$

of individuals in each compartment. As the model focuses on the dynamics of the epidemic over a short period (1000 days), we neglect demographic changes over this period and assume that the population size $N(r, a)$ in each location and age group is approximately constant, that is

$$S_t(r, a) + E_t(r, a) + I_t(r, a) + A_t(r, a) + R_t(r, a) + D_t(r, a) = N(r, a)$$

is constant.

## 2.2. A metapopulation SEIAR model

When each subpopulation $(r, a)$ is large and homogeneous, the dynamics of state variables may be described through the following system of equations, represented in figure 1:

$$
\left.
\begin{aligned}
\dot{S}_t(r, a) &= -\lambda_t(r, a)\, S_t(r, a), \\
\dot{E}_t(r, a) &= \lambda_t(r, a)\, S_t(r, a) - \beta E_t(r, a), \\
\dot{I}_t(r, a) &= p_a \beta E_t(r, a) - \gamma I_t(r, a), \\
\dot{A}_t(r, a) &= (1 - p_a)\beta E_t(r, a) - \gamma A_t(r, a), \\
\dot{D}_t(r, a) &= \gamma f_a I_t(r, a), \\
\dot{R}_t(r, a) &= \gamma(1 - f_a)I_t(r, a) + \gamma A_t(r, a) \\
N(r, a) &= S_t(r, a) + A_t(r, a) + E_t(r, a) + I_t(r, a) + R_t(r, a) + D_t(r, a),
\end{aligned}
\right\}
\tag{2.1}
$$

and

where

— $\beta$ is the incubation rate, and $1/\beta$ is the average incubation period;

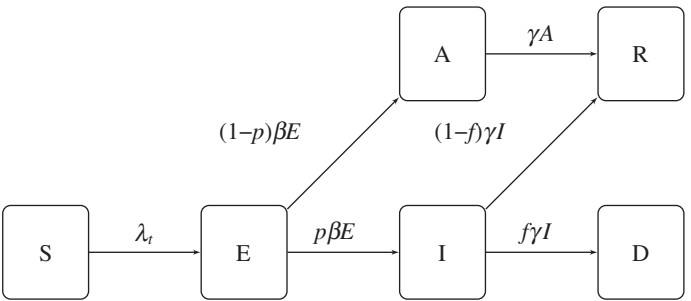

**Figure 1.** Epidemic dynamics.

— $\gamma$ is the rate at which infectious individuals recover;
— $0 < p_a < 1$ is the probability for an infected individual in age group $a$ to develop symptoms;
— $f_a$ is the infection fatality rate for age group $a$, representing the probability that an infected individual in age group $a$ dies from the disease; and
— the *force of infection* $\lambda_t(r, a)$, which measures the rate of exposure at location $r$ for age group $a$, is given by

$$\lambda_t(r, a) = \sum_{a' \notin \mathcal{W}} \sigma^r_{a,a'}(t) \frac{\alpha_1(a')\kappa I_t(r, a') + \alpha_0(a') A_t(r, a')}{N(r, a')}$$
$$+ \sum_{a' \in \mathcal{W}} \sigma^r_{a,a'}(t) \sum_{r'=1}^{K} M_{r,r'}(t) \frac{\alpha_1(a')\kappa I_t(r', a') + \alpha_0(a') A_t(r', a')}{N(r', a')}, \quad (2.2)$$

where $0 < \alpha_1(a) < 1$ (resp. $\alpha_0(a)$) is the infection rate per contact, i.e. the probability of infection conditional on contact for symptomatic (resp. asymptomatic) with individuals in age group $a$.

In the absence of reliable data on asymptomatic carriers, it is difficult to estimate $\alpha_0$. We will use as baseline model the case $\alpha_1 = \alpha_0 = \alpha$, and examine the impact of heterogeneous infection rates $\alpha_1(a) > \alpha_0(a)$ in §§3 and 5.

The force of infection in each subpopulation $(r, a)$ depends on the rate of contact with (infected) individuals in other subpopulations, which differentiates this model from a homogeneous model. These interactions occur through:

— *Contacts across age groups in the same region*: the term $\sigma^r_{a,a'}(t)$ represents the average number of persons from age class $a'$ encountered per day by a person from age class $a$ in region $r$ on a day $t$. For infectious individuals with symptoms, we assume a lower contact rate $\kappa\sigma < \sigma$ due to (partial) self-isolation (quarantine effect). This leads to the first term in (2.2).
— *Inter-regional mobility*: the second term in (2.2) corresponds to contacts between individuals in region $r$ and age class $a$ and those in the working population (age classes $a' \in \mathcal{W}$) commuting from other regions $r' \neq r$. $M_{r,r'}(t)$ represents the proportion of individuals from region $r'$ among the population of adults at a location $r$ at date $t$.

## 2.3. Stochastic dynamics

The deterministic dynamics (2.1) ignores the variability of outcomes [24] due to random factors not taken into account in the model. To account for this variability of outcomes, we model the variables $(S(t), E(t), I(t), A(t))$ as a continuous-time Markov point process [21,25] defined through its transition rates conditional on the history $\mathcal{H}_t$ up to date $t$

$$\mathbb{P}(\Delta S_t(r, a) = -1 | \mathcal{H}_t) = -\lambda_t(r, a) S_t(r, a)\Delta t + o(\Delta t)$$
$$\mathbb{P}(\Delta E_t(r, a) = 1 | \mathcal{H}_t) = \lambda_t(r, a) S_t(r, a)\Delta t + o(\Delta t)$$
$$\mathbb{P}(\Delta E_t(r, a) = -1 | \mathcal{H}_t) = \beta E_t(r, a)\Delta t + o(\Delta t)$$
$$\mathbb{P}(\Delta I_t(r, a) = 1 | \mathcal{H}_t) = p_a \beta E_t(r, a)\Delta t + o(\Delta t)$$
$$\mathbb{P}(\Delta I_t(r, a) = -1 | \mathcal{H}_t) = \gamma I_t(r, a)\Delta t + o(\Delta t)$$
$$\mathbb{P}(\Delta A_t(r, a) = +1 | \mathcal{H}_t) = (1 - p_a)\beta E_t(r, a)\Delta t + o(\Delta t)$$
$$\mathbb{P}(\Delta A_t(r, a) = -1 | \mathcal{H}_t) = \gamma A_t(r, a)\Delta t + o(\Delta t)$$
and
$$\mathbb{P}(\Delta D_t(r, a) = 1 | \mathcal{H}_t) = f_a \gamma I_t(r, a)\Delta t + o(\Delta t) \quad (2.3)$$

The stochastic dynamics (2.3) are consistent with the deterministic dynamics of (2.1) for large populations, in the sense that the population fractions represented by each compartment converge to those represented by the solution of (2.1) as $\min_r N(r)$ increases. However, even when the overall population is large, the stochastic dynamics (2.3) can substantially deviate from the deterministic model (2.1), especially in small subpopulations and in the early phases of the epidemic when the number of infected individuals in each region may be small, leading to random flare-ups and breakouts not present in the deterministic model. In the sequel, we use the stochastic model (2.3) for the dynamics of the state variables.

## 2.4. Policies for epidemic control

Social distancing policies (and lockdowns) affect epidemic dynamics by influencing (lowering) the social contact rates $\sigma_{ij}^r$ and the inter-regional mobility $M_{r,r'}$. To discuss targeted policies which may influence differently social contact rates at different locations, we decompose the baseline social contact matrix $\sigma^r$ as

$$\sigma^r(0) = \sigma^{r,H} + \sigma^{r,W} + \sigma^{r,S} + \sigma^{r,O}, \tag{2.4}$$

where the components correspond, respectively, to contacts at home ($\sigma^{r,H}$), work ($\sigma^{r,W}$), school ($\sigma^{r,S}$) and other locations ($\sigma^{r,O}$). Social distancing policies are then parametrized in terms of their impact on various components of the social contact matrix

$$\sigma_{ij}^r(t) = u_{ij}^{r,H}(t)\sigma_{ij}^{r,H} + u_{ij}^{r,S}(t)\sigma_{ij}^{r,S} + u_{ij}^{r,W}(t)\sigma_{ij}^{r,W} + u_{ij}^{r,O}(t)\sigma_{ij}^{r,O} \leq \sigma_{ij}^r(0), \tag{2.5}$$

where $0 \leq u_{ij}^{r,X}(t) \leq 1$ are modulating factors which measure the impact of the policy on social contacts between age groups $i$ and $j$ at a location $X$ in region $r$. In the absence of social distancing or confinement measures, we have $u_{ij}^{r,X}(t) = 1$; the value of $u_{ij}^{r,X}(t)$ reflects the fraction of social contacts between age groups $i$ and $j$ at location $X$ in region $r$ when the policy is applied.

This parametrization allows us to consider policies targeted towards subpopulation or specific regions. For example, school closure in region $r$ during time period $[t_1, t_2]$ corresponds to setting $u_{ij}^{r,S}(t) = 0$ for $t \in [t_1, t_2]$, while $0 < u_{ij}^{r,S} < 1$ corresponds to social distancing in schools, with lower values of $u_{ij}^{r,S}$ corresponding to stricter enforcement of measures.

In most cases, $u_{ij}^{r,X}(t)$ does not explicitly depend on the age groups $i, j$, as it is infeasible to discriminate between age groups when implementing social distancing requirements. Dependence on age groups arises when certain types of contacts are primarily related to certain age groups:

— Shielding of elderly populations: such policies affect the contact rates between elderly populations and other age groups.
— Work restrictions, which affect contacts between age groups of the working population (denoted $\mathcal{W}$): $u_{ij}^{r,W}(t) = u^{r,W}\mathbb{1}_{i \in \mathcal{W}}\mathbb{1}_{j \in \mathcal{W}}$.

Regarding the inter-regional mobility matrix $M$, following the interpretation discussed in §3.2, we modulate its value according to the fraction $u^{r,W}$ of the population who continue to commute, that is

$$M_{r,r'}(t) = u^{r,W}(t)M_{r,r'} + (1 - u^{r,W}(t))\mathbf{I}.$$

Here, $M_{r,r'}$ is the fraction of population in region $r$ whose habitual residence is in region $r'$.

The modulating factors $u_{ij}^{r,X}$ may be chosen in advance or expressed as a function of the state of the system. We distinguish:

— *pre-planned* (also called 'open-loop') policies, in which target values of modulating factors $u_{ij}^{r,X}(t)$ are decided in advance; and
— *adaptive* policies (also called 'closed loop' or feedback control), in which actions are decided and updated as a function of observed quantities such as number of daily reported cases or number of daily fatalities. This is similar to the regional 'tier' system adopted in England.

### 2.4.1. Comparative analysis of mitigation policies

To perform comparative analysis across different policies, we need to evaluate policy outcomes across two dimensions: health outcome and socio-economic impact.

We quantify the health outcome of each policy by the total number of fatalities during a reference period, taken to be $t_{max} = 1000$ days after the reference date of 1 March 2020. The length of this reference period is chosen such that it takes into account an eventual 'second wave' of fatalities. We denote this outcome by $D_{t_{max}}(u)$, which represent the total fatalities at date $t_{max}$ associated with policy $u$.

To quantify the socio-economic impact of a policy, we use as metric the reduction in social contact resulting from the policy over the horizon $[0, t_{max}]$, that is

$$J(u) = \sum_{t=1}^{t_{max}} \sum_{r=1}^{K} \sum_{i,j=1}^{M} \left( \sigma_{ij}^r(0) - \sigma_{ij}^r(t) \right) N(r, i), \tag{2.6}$$

defined in terms of person×day units.

The range of policies examined below lead to different outcomes in terms of fatalities $D_{t_{max}}(u)$ and social cost $J(u)$. A policy $v$ *dominates* (or improves upon) a policy $u$ if it leads to a similar or better health outcome at an equal or lower cost,

$$J(v) \leq J(u) \quad \text{and} \quad D_{t_{max}}(v) \leq D_{t_{max}}(u),$$

with at least one inequality being strict. A policy $u$ is *efficient* among a class of policies $U$ if it cannot be improved upon by any policy in this class. Given a set of policies $U$, the subset of efficient policies forms the *efficient frontier* of $U$.

Some recent economic models [6,9,10] formulate the trade-off in different terms, by introducing a concept of monetary *value of human life* in order to build a (monetary) welfare function combining both terms. Aside from ethical issues linked to the very concept of monetization of human life, there is no consensus on its actual value, which is a key determinant of the trade-off in this approach. Our approach avoids specifying such a value and aims at identifying the range of efficient policies, leaving the final choice of the trade-off to policy-makers.

In what follows, the goal is to determine the set of efficient policies and describe the characteristics and outcomes of such policies. Pre-planned policies are discussed in §§5.1 and 5.2, while adaptive policies are discussed in §6.

# 3. Data sources and parameter estimation

We now describes the model inputs as well as the methodology used in the parameter estimation. Table 1 contains a summary of model parameters.

## 3.1. Data sources

The basic inputs of the model are panel data on number of cases and fatalities reported at the level of Upper Tier Local Authorities (UTLA) in England, provided by the Public Health England and NHSX [36]. This defines the geographical granularity of the model: we partition the population of England into 133 regions as defined by the Nomenclature of Territorial Units for Statistics at level 3 (NUTS-3) [37].

For the purpose of our study, we distinguish $M = 16$ age groups, as shown in table 2, which is the maximum granularity allowed by the available estimates of age-dependent social contact rates and fatality rates. The size $N(r, a)$ of age group $a$ in region $r$ is retrieved using the population dataset provided by Eurostat [38]. Appendix A provides the list of UK regions used in this study and outlines the performed mapping procedure from UTLA to NUTS-3 regions to ensure consistency across data sources.

## 3.2. Modelling of inter-regional mobility

For our baseline estimate of inter-regional mobility, we use the 2011 Census data on location of usual residence and place of work in the UK, provided by the Office for National Statistics [35]. The dataset classifies people aged 16 and over in employment during March 2011 and shows the movement between their area of residence and workplace, defined in Local Administrative Units at level 1 (LAU-1) terms. We then map this data onto NUTS-3 regions using the lookup table between LAU-1 and NUTS-3 areas provided by the Office for National Statistics [39].

The data are then represented in the model through the inter-regional mobility matrix $M$, whose elements $M_{r,j}$ represent the fraction of population in region $r$ whose habitual residence is in region $j$. Denote by $\Pi(r, j)$ the population with residence registered in region $j$ and workplace registered in

**Table 1.** Summary of parameters for the COVID-19 model.

| model parameter name | symbol | value | source |
|---|---|---|---|
| infection rate | $\alpha$ | 0.055 (0.051, 0.062) | [3,26] |
| incubation rate | $\beta$ | 0.2 | [1,27,28] |
| recovery rate | $\gamma$ | 0.1 | [29–31] |
| infection fatality rate | $f$ | see table 5 | [32] |
| symptomatic ratios (low estimate) | $p_{\text{low}}$ | table 4 | [33] |
| symptomatic ratios (high estimate) | $p_{\text{high}}$ | table 4 | [12] |
| social contact matrix | $\sigma$ | appendix B | [34] |
| symptomatic contact adjustment | $\kappa$ | 0.5 | |
| regional adjustment for contact rates (pre-lockdown) | $d_r$ | figure 2 | |
| regional adjustment for contact rates (pre-lockdown) | $l_r$ | table 7 | |
| inter-regional mobility matrix | $M$ | | [35] |

**Table 2.** Age group distribution for England, 2019. Source: Eurostat [38].

| age group | [0,5) | [5,10) | [10,15) | [15,20) | [20,25) | [25,30) | [30,35) | [35,40) |
|---|---|---|---|---|---|---|---|---|
| size (millions) | 3.3 | 3.5 | 3.3 | 3.1 | 3.5 | 3.8 | 3.8 | 3.7 |
| fraction | 5.9% | 6.3% | 5.9% | 5.5% | 6.2% | 6.8% | 6.8% | 6.6% |
| age group | [40,45) | [45,50) | [50,55) | [55,60) | [60,65) | [65,70) | [70,75) | [75,100) |
| size (millions) | 3.4 | 3.8 | 3.9 | 3.6 | 3.1 | 2.8 | 2.8 | 4.7 |
| fraction | 6.0% | 6.7% | 7.0% | 6.5% | 5.5% | 5.0% | 4.9% | 8.4% |

region $r$ for $r \neq j$. In addition, we denote by $\Pi(r, r) = \sum_{a \in \mathcal{W}} N(a, r)$, where $\mathcal{W} = \{5, 6, \ldots, 12\}$ represents the total population at location $r$ in the age category [20, 60) years. Then, we estimate the coefficients of $M_{r,j}$ by

$$\widehat{M}_{r,j} = \frac{\Pi(r, j)}{\sum_{i=1}^{K} \Pi(r, i)}. \tag{3.1}$$

## 3.3. Epidemiological parameters

Epidemiological parameters were either estimated from publicly available sources [40,41] or set to values consistent with recent clinical and epidemiological studies in COVID-19 [26,32,42].

### 3.3.1. Social contact rates

Contact rates across age classes have been estimated in studies by Mossong *et al.* [34,41] and Béraud *et al.* [43]. We use the estimates of social contact rates provided by Mossong *et al.* [34] for the 16 age groups defined in table 2. Using the PyRoss methodology [40], we further decompose the contact matrix, as in (2.4), into four components representing contacts at home ($\sigma^H$), work ($\sigma^W$), school ($\sigma^S$) and other locations ($\sigma^O$). Estimation methods and parameter values for these matrices are discussed in appendix B.

Contact rates may vary across different regions due to the heterogeneity in socio-economic composition structure and specific regional characteristics, such as population density, level of urbanization and the level of use of public transport. To account for this heterogeneity, we parametrize the (pre-lockdown) contact matrix in region $r$ as $\sigma^r(0) = d_r \sigma$ where the regional adjustment factors $\{d_r : r = 1, \ldots, 133\}$ are estimated to reproduce the regional growth rate of reported cases before the lockdown period. The results are displayed in figure 2. Table 3 provides a summary of selected characteristics of five regions with the highest values of the regional adjustment factors $d_r$.

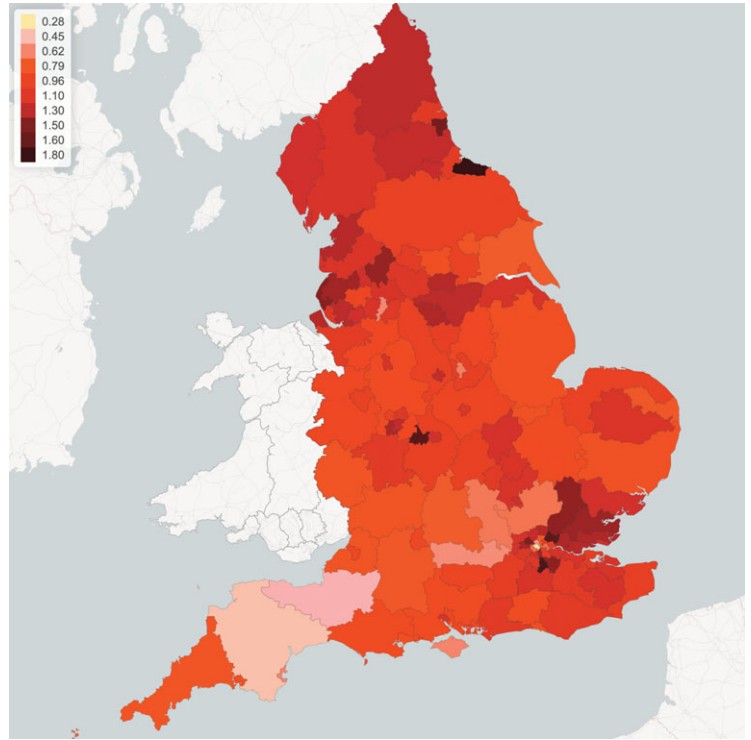

**Figure 2.** Regional multiplier $d_r$ for social contact matrix, implied by epidemic dynamics pre-lockdown (before 23 March 2020).

**Table 3.** Summary of regions with the highest regional multiplier $d_r$ for social contact matrix. Number in brackets signify the respective rank of the measured quantity.

| region | $d_r$ | density | inward mobility | outward mobility | population |
|---|---|---|---|---|---|
| UKC12 | 1.80 | 925.9 (59) | 17.6% (82) | 19.1% (105) | 276 988 (102) |
| UKI62 | 1.68 | 4518.4 (67) | 16.6% (87) | 43.2% (12) | 389 473 (59) |
| UKG32 | 1.64 | 1205.5 (59) | 14.5% (96) | 46.2% (6) | 215 055 (120) |
| UKI53 | 1.62 | 6161.9 (11) | 23.4% (50) | 41.6% (16) | 587 575 (25) |
| UKC23 | 1.52 | 2026.9 (49) | 13.7% (103) | 24.0% (78) | 277 733 (99) |

As seen in figure 2, our findings imply heterogeneity of social contact rates across regions. As we will observe below, these differences have a considerable impact on regional epidemic dynamics.

### 3.3.2. Incubation rate

Following the study of Ferguson *et al.* [1], we use an incubation rate $\beta = 0.2$, which corresponds to an incubation period of approximately 5 days. This is further supported by several empirical studies on diagnosed cases in China outside Hubei province. An early study of Backer *et al.* [44] based on 88 confirmed cases, which uses data on known travel to and from Wuhan to estimate the exposure interval, indicates a mean incubation period of 6.4 days with a 95% confidence interval (CI) of 5.6–7.7 days. Linton *et al.* [27], based on 158 confirmed cases, estimate a median incubation period of 5.0 days with 95% CI of 4.4–5.6 days and estimate the incubation period to have a mean of around 5 days with 95% CI of 4.2–6.0 days. Lauer *et al.* [28] estimates a median of incubation period to be 5.1 days with 95% CI of 4.5–5.8 days, based on 181 cases over the period of 4 January to 24 February 2020.

### 3.3.3. Proportion of symptomatic and asymptomatic infections

The probability $p$ that an infected individual develops symptoms is an important parameter for epidemic dynamics, yet subject to a high degree of uncertainty: studies on various datasets [12,33,45–47] are based

**Table 4.** Age-dependent symptomatic ratios, $p$. Source: Office for National Statistics [33] and Davies *et al.* [12].

| age group | [0,5) | [5,10) | [10,15) | [15,20) | [20,25) | [25,30) | [30,35) | [35,40) |
|---|---|---|---|---|---|---|---|---|
| $p_{low}$ | 0.075 | 0.075 | 0.05 | 0.05 | 0.15 | 0.15 | 0.21 | 0.21 |
| $p_{high}$ | 0.15 | 0.15 | 0.1 | 0.1 | 0.3 | 0.3 | 0.42 | 0.42 |
| age group | [40,45) | [45,50) | [50,55) | [55,60) | [60,65) | [65,70) | [70,75) | [75,100) |
| $p_{low}$ | 0.23 | 0.23 | 0.28 | 0.28 | 0.41 | 0.41 | 0.375 | 0.375 |
| $p_{high}$ | 0.45 | 0.45 | 0.56 | 0.56 | 0.82 | 0.82 | 0.75 | 0.75 |

**Table 5.** Age-dependent infection fatality rates. Source: Verity *et al.* [32].

| age group | [0,5) | [5,10) | [10,15) | [15,20) | [20,25) | [25,30) | [30,35) | [35,40) |
|---|---|---|---|---|---|---|---|---|
| $f$ (%) | 0.002 | 0.002 | 0.01 | 0.01 | 0.05 | 0.05 | 0.1 | 0.1 |
| age group | [40,45) | [45,50) | [50,55) | [55,60) | [60,65) | [65,70) | [70,75) | [75,100) |
| $f$ (%) | 0.2 | 0.2 | 0.6 | 0.6 | 2.00 | 2.00 | 4.0 | 7.5 |

on small samples and yield a wide range of estimates. In particular, an early estimate from the Diamond Princess cruise ship [47] and Japanese evacuation flights from Wuhan yielded estimates as high as $p \simeq 0.7 - 0.8$ [48], while a July 2020 study by the Office for National Statistics [33], based on a much larger sample, showed that $p$ can be as low as 0.23. However, clinical studies [12] indicate that this probability may strongly depend on the age group considered.

We use a range of values for the age-dependent probability $p_a$ whose upper bound is consistent with Davies *et al.* [12] and whose lower bound is consistent with the estimates provided by the Office for National Statistics [33]. These values are displayed in table 4. Given the much larger sample size used in the study of Office for National Statistics [33], we use the corresponding estimates (low values, denoted as $p_{low}$ in table 4) as benchmark unless stated otherwise.

### 3.3.4. Recovery rate $\gamma$

In line with Cao *et al.* [29], Li *et al.* [30] and Rocklöv *et al.* [31], we use a recovery rate $\gamma = 0.1$, which corresponds to an average infectious period of 10 days.

### 3.3.5. Infection fatality rates

We denote by $f_a$ the (infection) fatality rate for age group $a$. In practice, these parameters are difficult to estimate during outbreaks and estimates may be subject to various biases [49]. Note that the *infection* fatality rate (IFR) is different from (and generally much smaller than) the *case* fatality rate.

Fatality rates for COVID-19 have been observed to be highly variable across age groups [32,42,50]. Based on the infection fatality rates provided in Verity *et al.* [32] for different age groups and the UK population distribution, we derive the aggregated IFR for the respective 16 age groups of interest as summarized in table 5. These estimates are consistent with data obtained from other countries; for example, see Salje *et al.* [50].

## 3.4. Estimation of the infection rate

We use a simulation-based indirect inference method [51] for estimating the infection rates $\alpha_0$, $\alpha_1$ for asymptomatic and symptomatic carriers across age classes.

Due to the lack of direct observability of asymptomatic carriers and lack of granularity of case data (the breakup by age class in each region is not reported), we first consider the case where $\alpha_0(a) = \alpha_1(a) = \alpha$, and then explain how to adjust for asymptomatic/symptomatic carriers and age-dependence.

**Table 6.** Age-dependent infection rates: symptomatic ($\alpha_1$) versus asymptomatic ($\alpha_0$).

| age group | [0,5) | [5,10) | [10,15) | [15,20) | [20,25) | [25,30) | [30,35) | [35,40) |
|---|---|---|---|---|---|---|---|---|
| $\alpha_0$ | 0.045 | 0.045 | 0.048 | 0.048 | 0.039 | 0.039 | 0.034 | 0.034 |
| $\alpha_1$ | 0.174 | 0.174 | 0.185 | 0.185 | 0.148 | 0.148 | 0.132 | 0.132 |
| age group | [40,45) | [45,50) | [50,55) | [55,60) | [60,65) | [65,70) | [70,75) | [75,100) |
| $\alpha_0$ | 0.033 | 0.033 | 0.031 | 0.031 | 0.025 | 0.025 | 0.027 | 0.027 |
| $\alpha_1$ | 0.128 | 0.128 | 0.118 | 0.118 | 0.098 | 0.098 | 0.102 | 0.102 |

**Table 7.** Estimated values for regional adjustments $d_r$ and $l_r$ in NUTS-1 regions.

| NUTS-1 region | pre-lockdown ($d_r$) | lockdown ($l_r$) |
|---|---|---|
| South West (UKK) | 0.729 | 0.099 |
| East Midlands (UKF) | 0.952 | 0.134 |
| London (UKI) | 1.143 | 0.100 |
| West Midlands (UKG) | 1.020 | 0.126 |
| Yorkshire and Humber (UKE) | 1.069 | 0.137 |
| South East (UKJ) | 0.920 | 0.116 |
| North East (UKC) | 1.260 | 0.131 |
| North West (UKD) | 1.122 | 0.137 |
| East of England (UKH) | 0.994 | 0.129 |

To estimate $\alpha$, we simulate the stochastic model (2.3) for a range of values $0.03 \leq \alpha \leq 0.15$. The value of $\alpha$ is estimated by matching the logarithmic growth rates of the simulated reported cases with that of reported cases $C_t$ in England.

For the simulation, we use parameters specified in table 1 and the following initial conditions for $t_0 =$ 10 March 2020,

$$E_{t_0}(r, a) = \frac{N(r, a)}{\sum_{a' \in \mathcal{W}} N(r, a')} \frac{C_{t_0+5}(r)}{p_2 \, \pi} \, 1_{a \in \mathcal{W}} \tag{3.2}$$

where $\mathcal{W} = \{5, 6, 7, 8, 9, 10, 11, 12\}$ corresponds to age groups in the working population and $A_{t_0}(r, a) = 0$, $D_{t_0}(r, a) = 0$, $I_{t_0}(r, a) = 0$ for all $a$. These initial conditions ensure that the simulations agree on average with regional case numbers on 15 March 2020, for all values of $\alpha$.

This procedure yields an estimated value of $\widehat{\alpha} = 0.055$ and a confidence interval $[0.051, 0.062]$. This value of $\widehat{\alpha}$, together with the model parameters in table 1, yields a good fit of the pre-lockdown evolution of case numbers.

These results are consistent with estimates obtained in Donnat & Holmes [3] and Dorigatti et al. [26] using data from other countries.

The above estimate of $\alpha$ represents an average infection rate. Recent epidemiological evidence suggests that symptomatic carriers in a given age group $a$ have a higher rate of infection $\alpha_1(a)$ than asymptomatic carriers (whose infection rate is denoted $\alpha_0(a)$) [52]. Sayampanathan et al. [52] estimate that when adjusted for age and gender, the incidence of COVID-19 among close contacts of a symptomatic index case was 3.85 times higher than for close contacts of an asymptomatic carrier, that is $\alpha_1(a) \simeq 3.85 \, \alpha_0(a)$.

Assuming the average infection rate is identical across age groups, we obtain

$$\alpha = p_a \alpha_1(a) + (1 - p_a) \alpha_0(a).$$

These two constraints lead to unique values $(\alpha_0(a), \alpha_1(a)$, shown in table 6) consistent with $p_a$ and global estimate for $\alpha$.

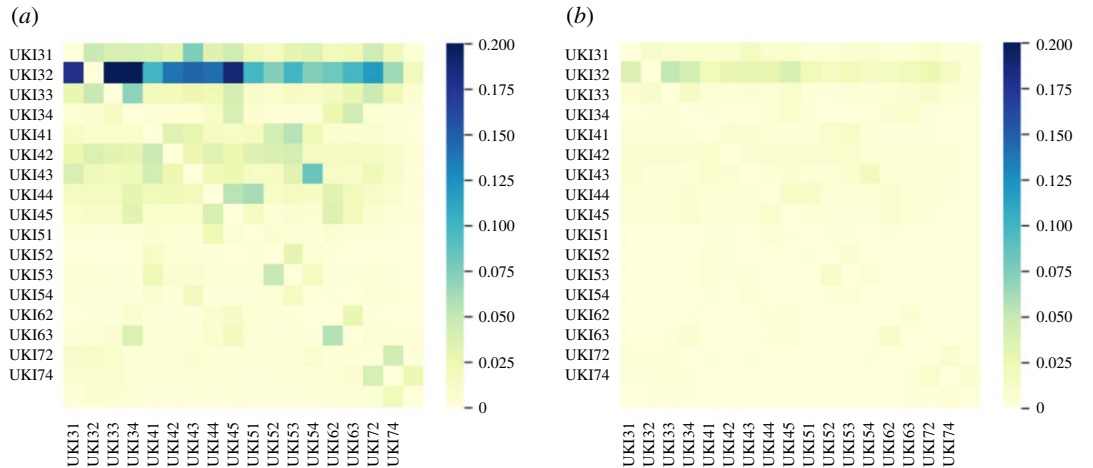

**Figure 3.** Inter-regional mobility across London boroughs. (*a*) Pre-lockdown: before 23 March 2020. (*b*) During lockdown: 23 March–10 June 2020.

## 3.5. Inter-regional mobility and social contact during confinement

Confinement measures were implemented across the UK starting 23 March 2020 via the Coronavirus Act.[2] During this 'lockdown' period schools and workplaces were closed and social contact was reduced, as evidenced by mobility data.[3] However, mobility data also reveal regional differences in the impact of the lockdown.

We model the reduction in inter-regional mobility through an adjusted mobility matrix

$$\widehat{M}_{r,r'}(t) = q\,\widehat{M}_{r,r'} + (1-q)I, \quad \text{where } 0 < q < 1, \tag{3.3}$$

and $\widehat{M}_{r,r'}$ is the inter-regional mobility matrix defined in (3.1). According to the Labour Force Survey data from 2018/19 [53], 7.1 million adults across the UK are considered as 'key workers'. We set $q = 20\%$ to take into account the fact that these key workers continued to access their workplace during the lockdown period. This is also consistent with the methodology in Rawson *et al.* [7] and empirical studies of Santana *et al.* [54] on mobility changes before and after lockdown in the UK. Figure 3 shows the submatrix corresponding to daily mobility across London boroughs, and illustrates the observed dramatic drop in commute patterns.

We model the impact of confinement on the social contact matrix through a regional multiplier $l_r$

$$\sigma^r(t) = l_r \times \sigma(0), \tag{3.4}$$

where $l_r \leq d_r$ represents the reduction in social contacts during the lockdown period; $l_r = d_r$ corresponds to the pre-lockdown level of social contact. The value of $l_r$ is estimated from panel data on regional epidemic dynamics during the period from 23 March to 1 June 2020, using a least-squares logarithmic regression on the number of observed regional cases (see table 7).

The average value of this reduction factor is found to be

$$\frac{\sum_{r=1}^{133} N(r)l_r}{\sum_{r=1}^{133} N(r)} = 0.12,$$

which is an average reduction of 88% in social contacts, an order of magnitude corroborated by mobility data [54], showing that the lockdown was very effective in reducing social contacts.

## 3.6. Goodness-of-fit

Having estimated the model parameters using data on reported cases between 10 March and 20 May 2020 we assess the goodness-of-fit and out-of-sample performance using reported cases and fatalities between 21 May and 22 June 2020. Figures 4 and 5 show that the model is able to reproduce the in-sample and out-of-sample evolution of case numbers and fatalities, at national level as well as regional level.

[2]See https://www.legislation.gov.uk/ukpga/2020/7/contents/enacted.

[3]See https://www.oxford-covid-19.com/.

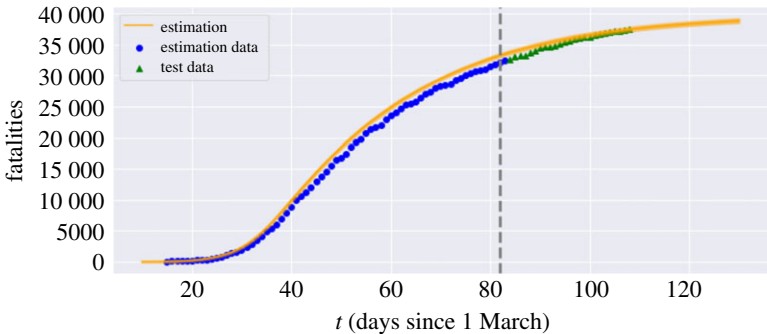

**Figure 4.** Fatalities in England: comparison of model with data. Grey dashed line: separation between estimation sample and test data; orange line: model simulation; blue dot: in-sample data; green triangle: out-of-sample data.

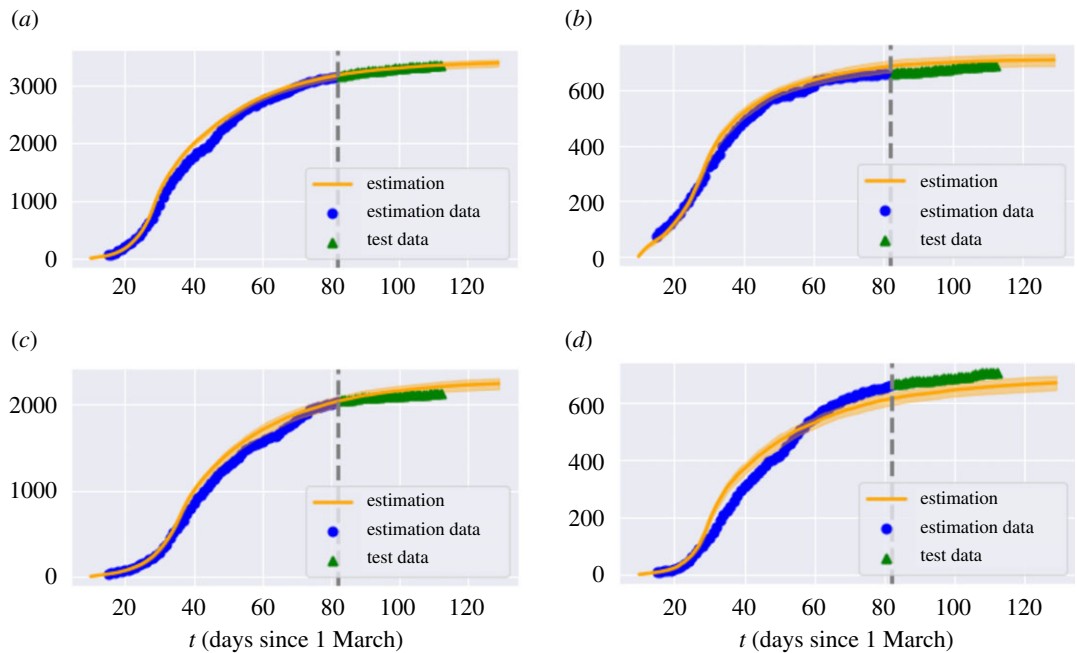

**Figure 5.** Cumulative reported cases in selected regions. Grey dashed line: separation between estimation sample and test data; orange line: average of 50 simulated scenarios; blue dot: in-sample data; green triangle: out-of-sample data. (*a*) Birmingham, (*b*) Westminster, (*c*) Oxfordshire, (*d*) West Essex.

# 4. Observable quantities and uncertainty

When applying such models to epidemic data, a key point is to realize that the state variables $S, E, I, A, R$ are not directly observed (and certainly not in real time) but need to be inferred from other observable quantities.

In the absence of widespread testing, public health authorities are faced with the problem of controlling a system under partial observation. This lack of direct observability has some implications for the estimation and interpretation of the model, which we briefly discuss here.

## 4.1. Observable quantities

The two main observables in COVID-19 data are

— the cumulative number of *reported* cases; and
— the cumulative number of COVID-19 fatalities $D_t$.

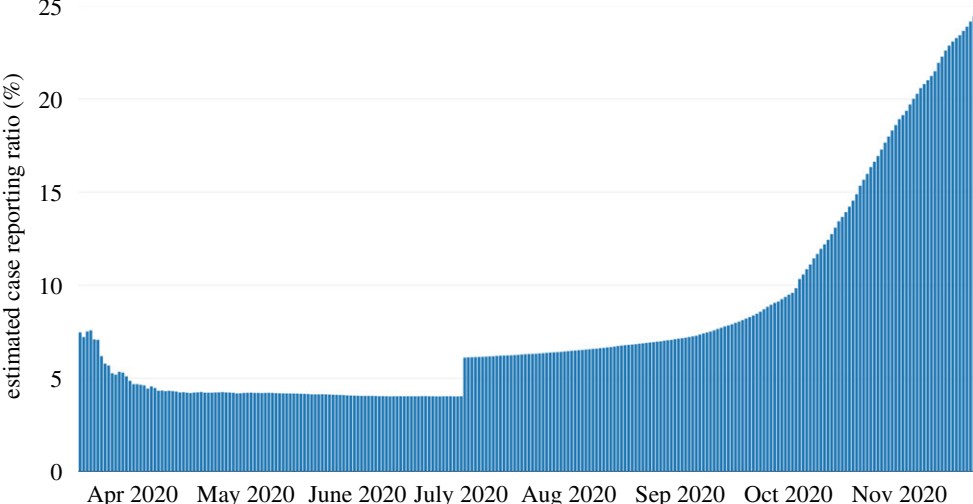

**Figure 6.** Estimate of case reporting probability $\pi(t)$ based on a comparison of fatalities and reported cases.

Of the two, fatalities are generally considered more reliable, as deaths are nearly always reported, while identification of cases requires testing or self-reporting. We thus identify the observed number of fatalities with the state variable $D_t$.

In the absence of widespread testing, only a fraction $\pi$ of cases are reported. This fraction may change with time due to testing campaigns.[4] We therefore cannot assume the number of infectious cases to be directly observed: rather, we estimate it from the fatality count $D_t$ (see also Jombart *et al.* [55]).

Let $C_t$ be the cumulative number of (symptomatic) infectious cases. Assuming that

— the daily number $r(t)$ of reported cases is a fraction $\pi(t)$ of new cases, that is

$$r(t) = \pi(t)(C_{t+1} - C_t); \tag{4.1}$$

— deaths occur on average $T$ days after detection;

we obtain that the daily fatality count is proportional to the lagged number of new cases,

$$D_{t+T+1} - D_t \simeq \overline{f}(C_{t+1} - C_t) = \frac{\overline{f}}{\pi(t)}\, r(t), \tag{4.2}$$

where $\overline{f}$ is the (average) infection fatality rate. We use these relations to obtain an estimate for the cumulative number $C_t$ of symptomatic infections and the reporting ratio $\pi(t)$.

Using equation (4.2), we estimate the average delay $T$ between case reporting and death by identifying the lag $T$ which maximizes the correlation between the $D_{t+T+1} - D_t$ and $r(t)$. Using an average fatality rate of $\overline{f} = 0.9\%$ for the UK as in [1] (see discussion in §3.3), we estimate the reporting probability to be

$$\widehat{\pi(t)} = \frac{\overline{f}\, r(t)}{D_{t+T+1} - D_t}, \tag{4.3}$$

which implies that the total number of cases in England is more than 20 times the reported number. As shown in figure 6, prior to June 2020 this reporting ratio was around $\widehat{\pi(t)} = 4.5\%$; with the subsequent increase in testing, the estimated reporting ratio has steadily increased to more than 20% in November 2020.

## 4.2. Implications of partial observability

A key issue in epidemic control is the availability of reliable indicators for the intensity of an ongoing epidemic. Public health authorities have communicated the daily number of reported cases and fatalities, and these have served as inputs for policy planning.

An important corollary of the above discussion is that, given the combination of random factors affecting dynamics and the considerable uncertainty on the actual number of new infections, it is perfectly possible to observe a run of many consecutive days without new reported cases while in fact the *actual* number of infections is on the rise.

---

[4]See https://ourworldindata.org/coronavirus-testing.

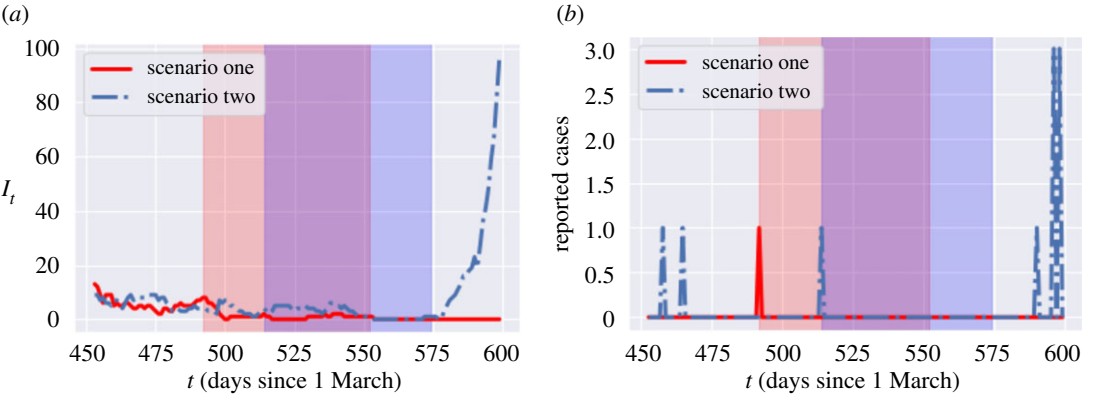

**Figure 7.** Example of latent progression of the epidemic with zero reported case for 60 consecutive days (red shaded area for scenario 1 and blue shaded area for scenario 2). Reporting probability is $\pi = 4.5\%$. (*a*) Number of infected $I_t$, (*b*) reported cases.

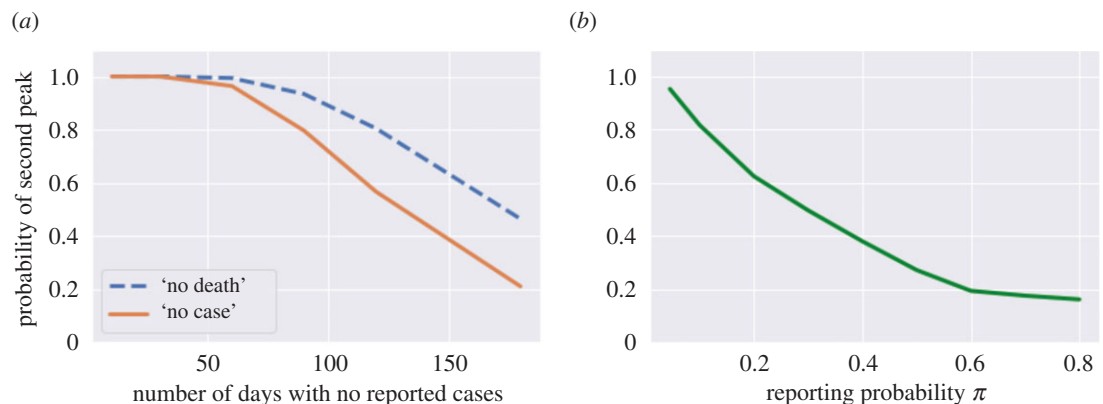

**Figure 8.** Probability of observing a second peak after a period with no cases reported. (*a*) Probability of having a second peak in infections after no reported cases (solid line) and no fatalities (dashed line) for $L$ consecutive days (low symptomatic ratios), (*b*) probability of having a second peak in infections following 60 consecutive days with no reported cases, as a function of reporting probability $\pi$.

Figure 7 shows an example of scenario in our model where, for 60 consecutive days, although a small number of (symptomatic and asymptomatic) cases appear, due to the low detection probability ($\pi = 4.5\%$), none of them is reported. Nevertheless, after a run of 60 days without any reported cases (blue shaded area in figure 7), which may prompt public health authorities to lower their guard, the epidemic takes off again. Figure 7 displays in fact two sample paths with the same initial conditions, which differ only through the stochasticity of the dynamics. The fact that the breakout occurs only in one of the two scenarios (in blue) but not in the other illustrates how random flare-ups may originate from a small group of undetected cases.

Figure 8*a* shows the probability of observing a second peak in infections when social distancing measures are lifted after no reported cases for $L$ consecutive days. This probability is estimated using 500 simulated paths from (2.3). It is striking to observe that, even after 60 days with no reported cases, the probability of observing a resurgence of the epidemic is around 40%. Figure 8*a* (blue dashed line) shows the same probability conditional on observing no fatalities for $L$ consecutive days.

These observations point to the importance of broader testing: as shown in figure 8*b*, an increase in the probability $\pi$ of detecting new cases leads to a strong decrease in the probability of misdiagnosing the end of the epidemic, as in the scenario described above.

# 5. Comparative analysis of epidemic control policies

## 5.1. Confinement followed by social distancing

We first consider the impact of a national 'lockdown' followed by social distancing, which reflects the situation in the UK between March 2020 and August 2020. We examine in particular the impact of a

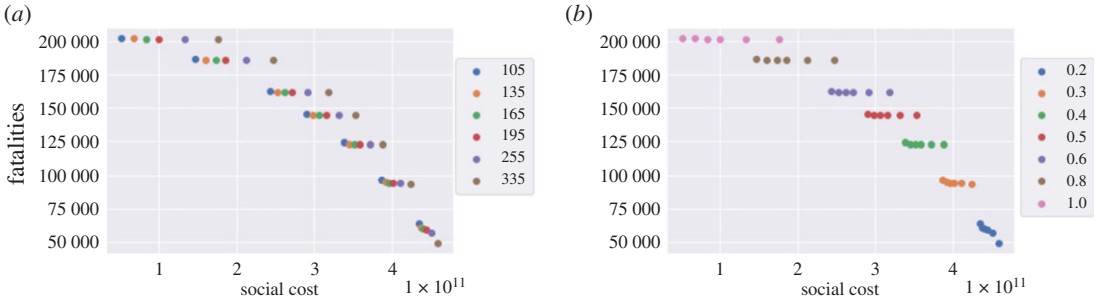

**Figure 9.** Fatalities against social cost for different $T$ and $m$ values (results for low symptomatic ratios). (a) Impact of lockdown duration $T$, (b) impact of compliance level $m$.

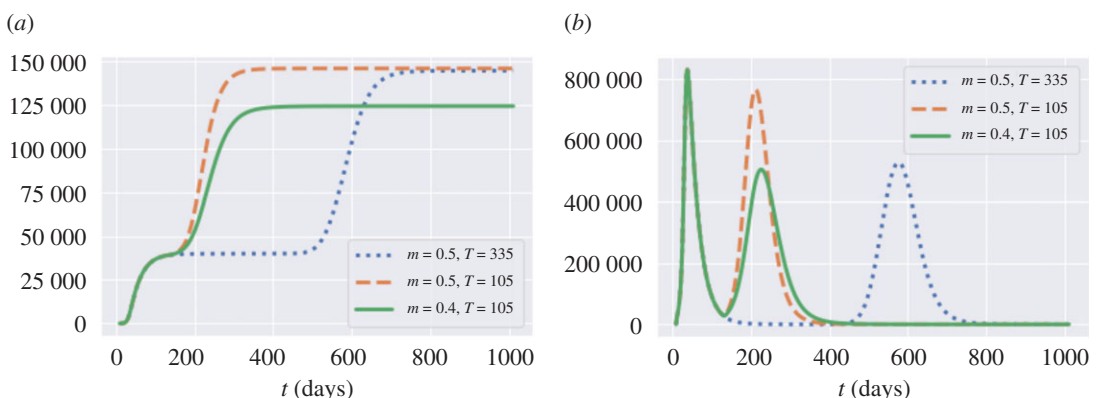

**Figure 10.** Comparison of three policies: blue dotted line: $m = 0.5$ and $T = 335$; orange dashed line: $m = 0.5$ and $T = 105$; green solid line: $m = 0.4$ and $T_2 = 105$. Average across 50 simulated scenarios. (a) Cumulative fatalities in England, (b) dynamics of $I_t$ in England.

lockdown duration $T$ and the level of social distancing after lockdown on the number of fatalities and the associated social cost. To do so, we parametrize the contact matrix as

$$\sigma^r(t) = \begin{cases} l_r\,\sigma & \text{for } t_0 \leq t \leq t_0 + T \quad \text{(lockdown),} \\ ((1-m)l_r + md_r)\,\sigma & \text{for } t > t_0 + T \quad \text{(after lockdown),} \end{cases} \quad (5.1)$$

where $l_r$ measures the level of social distancing under lockdown, as estimated from observations for the period from 23 March to 31 May, and the parameter $m \in [0, 1]$ measures the level of compliance with social distancing measures. A value of $m$ close to zero indicates a level of social contact similar to lockdown, while $m = 1$ corresponds to normal levels of social contact.

The origin date $t = 0$ corresponds to 1 March 2020. All scenario simulations include a lockdown starting at $t_0 = 23$ March 2020. We consider a range $105 \leq T \leq 335$ for the lockdown duration and $0.2 \leq m \leq 1$ for post-lockdown social distancing levels. Note that the actual duration of the first lockdown in England corresponded to $T = 105$.

As shown in figure 9a, the level of social distancing *after* the confinement period is observed to be more important (figure 9b) than the length of the confinement period (figure 9a). This is consistent with the findings in Lipton & Lopez de Prado [13]. Smaller values of $m$, associated with stricter social distancing, lead to a lower number of fatalities but for at an increased social cost (figure 9b). On the other hand, the lengthening of the lockdown duration $T$, while significantly increasing the associated social cost, does not result in a significant reduction in the number of fatalities, especially if social distancing is not respected after lockdown.

Figure 9 also shows that some of these policies are inefficient, in the sense that we can reduce fatalities *and* the social cost simultaneously by shortening the lockdown period or by relaxing social distancing constraints, as shown in figure 10.

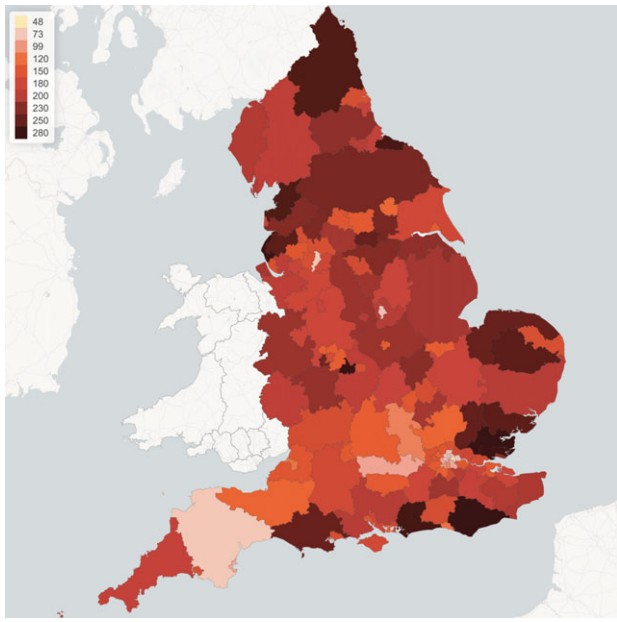

**Figure 11.** Lockdown of 105 days followed by social distancing ($m = 0.3$): regional mortality per 100 000 inhabitants.

**Table 8.** Outcomes for policies represented in figure 10.

| policy | blue dotted: | orange dash: | green solid: |
|---|---|---|---|
| | $m = 0.5$, $T = 335$ | $m = 0.5$, $T = 105$ | $m = 0.4$, $T = 105$ |
| social cost ($\times 10^{11}$) | 3.5 | 2.9 | 3.4 |
| projected fatalities | 144 600 | 146 000 | 124 400 |

As shown in table 8, by comparing the orange and blue plots in figure 10, which represent the same post-lockdown compliance level ($m = 0.5$), we observe that extending the lockdown duration increases social cost without reducing the total number of fatalities. On the other hand, comparing the orange and green plots, which correspond to the same lockdown duration of $T = 105$ days, shows that moving the compliance level from $m = 0.5$ to $m = 0.4$ reduces the second peak amplitude by 35% and fatalities by 13.9%.

### 5.1.1. Regional heterogeneity

While the policies discussed here are applied uniformly across all regions, we observe a significant heterogeneity in mortality levels across regions, as well as in terms of the timing and amplitude of a second peak in infections. As shown in figure 11, some regions exhibit mortality levels up to four times higher than others. This huge disparity in mortality rates cannot be explained by demographic differences alone, which are much less pronounced: more important seems to be the differences in social contact patterns, as illustrated in figure 2. Indeed, as shown in figure 12$a$, there is a positive correlation (above 40%) between regional COVID-19 mortality and the intensity of social contact as measured by the parameter $d_r$, defined in §3.3. Figure 12$b$ shows that this heterogeneity is also reflected in the timing and amplitude of second peaks.

### 5.2. Targeted policies

We now consider the impact of social distancing measures targeting particular age groups or environments (school, work, etc.) following a lockdown of duration $T$, by setting

$$\sigma_{ij}^r(t) = \begin{cases} l_r \ \sigma & \text{for} \quad t_0 \leq t \leq t_0 + T \quad \text{(lockdown)} \\ \sigma_{ij}^{r,H} + u_{ij}^S \sigma_{ij}^{r,S} + u_{ij}^W \sigma_{ij}^{r,W} + u_{ij}^O \sigma_{ij}^{r,O} & \text{for} \quad t > t_0 + T. \end{cases} \tag{5.2}$$

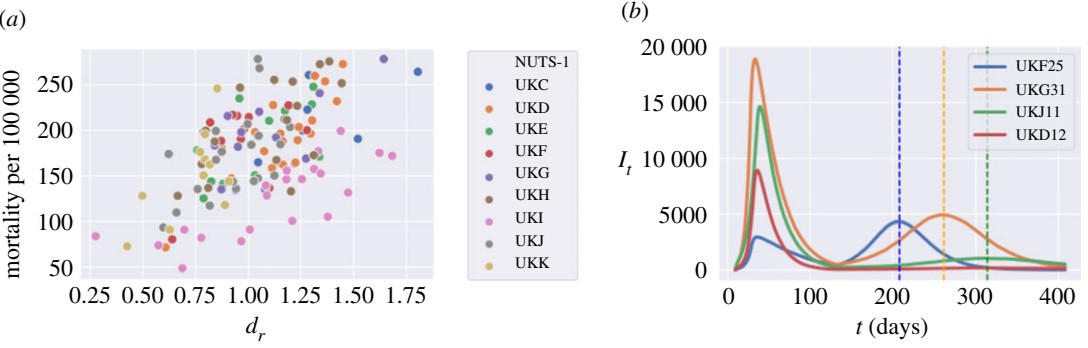

**Figure 12.** Regional outcomes for lockdown of 105 days followed by social distancing ($m = 0.3$). (a) Level of social contact ($d_r$) against COVID-19 mortality (per 100 000 inhabitants), (b) Regional dynamics of symptomatic infections ($I_t$): North Northamptonshire (UKF25), Birmingham (UKG31), Berkshire (UKJ11) and East Cumbria (UKD12). Dotted lines denote the second-peak times.

We consider different targeted measures after a lockdown period of $T = 105$ days (the actual duration of the lockdown in England): school closure, shielding of elderly populations and workplace restrictions, restrictions on social gatherings and combinations thereof. Note that there is no control over the social contacts at home.

### 5.2.1. School closures

Although most of the infected population below 20 is asymptomatic, they may in turn infect the population over 60 who are more likely to develop symptoms. School closure corresponds to $u^S = 0$, school reopening with social distancing correspond to $u^S = 0.5$, and school reopening without social distancing correspond to $u^S = 1$.

### 5.2.2. Shielding

The high infection fatality rates among elderly populations (age groups over 60) have naturally led to considering shielding policies for these populations. We model this as a reduction in social contacts of these age groups to the level observed under lockdown,

$$\sigma_{i,j}^r(t) = l_r(\sigma_{i,j}^H + \sigma_{i,j}^S + \sigma_{i,j}^W + \sigma_{i,j}^O) \quad \text{if} \quad i \in \{13, 14, 15, 16\} \quad \text{or } j \in \{13, 14, 15, 16\}.$$

### 5.2.3. Workplace restrictions

We model the impact of a restricted return to work after confinement by assuming different proportion of workforce return after the lockdown period by choosing

$$0.2 < u^W < 1 \quad \text{for } t > t_0 + T, \tag{5.3}$$

the lower bound $u^W = 0.2$ corresponding to restricting workplace return to 'essential workers', as discussed in §3.5. Since workplace restrictions have an effect on commuting, such measures also have an impact on the inter-regional mobility matrix

$$M_t = u^W(t)M_0 + (1 - u^W(t))I, \tag{5.4}$$

where $M_0(r, r')$ is the baseline mobility matrix defined in (3.1).

### 5.2.4. Restrictions on social gatherings

Although social activities, such as gatherings at pubs or sports events, may aggravate the contagion of COVID-19, keeping certain levels of social activities is important to the economic recovery and the well-being of individuals. The parameter $u^O$ measures the fraction of social gatherings: during the lockdown this fraction was estimated to be as low as 20% (see §3.5). In what follows, we consider $u^O \in [0.3, 1.0]$ after the period of lockdown.

**Table 9.** Impact of school closures and social distancing at schools: outcomes averaged across 50 simulated scenarios, $u^H = u^W = 1$, $u^O = 0.5$.

| | school closure $u^S = 0$ | social distancing at school $u^S = 0.5$ | normal school regime $u^S = 1.0$ |
| --- | --- | --- | --- |
| social cost ($10^{11}$) | 2.2 | 1.9 | 1.5 |
| projected fatalities | 153 900 | 157 000 | 159 300 |

### 5.2.5. Pubs and schools

Table 9 shows the impact of school closures and social distancing at schools on projected fatalities and social contacts. Reopening of schools, while reducing significantly the social cost, does not seem to lead to a significant increase in fatalities.

We compare two post-confinement policies, one (labelled as 'schools') consisting in leaving schools open while social gatherings are restricted ($u^S = 1$, $u^O = 0.2$), and the other (labelled as 'pubs') consisting in closing schools while not restricting social gatherings ($u^S = 0$, $u^O = 1$). The social cost for the 'pubs' policy is 2.3, while the cost for the 'schools' policy is 3.0. However, as shown in figure 14, the 'open school' policy leads to 35% fewer fatalities compared to the 'open pubs' policy.

### 5.2.6. Shielding of senior citizens

We have examined the impact of shielding in isolation and also in combination with other measures such as school closure and social distancing.

As shown in figure 13a, whether applied in isolation or in combination with other measures, shielding of elderly populations is by far the most effective measure for reducing the number of fatalities. As clearly shown in figure 13a, regardless of the trade-off between social cost and health outcome, a policy which neglects shielding of the elderly is not efficient and its outcomes can always be improved through shielding measures (see figures 13 and 15).

For policies without shielding, the level of social gatherings, $u^O$, is the leading factor to determine the efficiency frontier. In figure 16a, the efficiency frontier contains two classes of policies:

— 'School and work' policies, which do not include any restrictions on school or work ($u^S = 1$, $u^W = 1$) but varying level of restrictions on social gatherings ($0.3 \leq u^O \leq 1$). Within this class of policies, different level of social gatherings lead to very different outcome of fatalities, as illustrated in figure 16a.
— 'No pubs' policies, where social gatherings outside school and work are restricted ($u^O = 0.3$), with different levels of social distancing $u^S \in \{0, 0.5, 1\}$ $u^W \in [0.2, 1]$ at school and work.

However, as observed in figure 16b, these policies are not efficient when shielding measures are put in place for the elderly.

Under shielding, the spectrum of efficient policies is parametrized by the fraction $u^W$ of the workforce returning to work. As shown in figure 16c, we can distinguish two classes of efficient policies under shielding:

— 'School and pubs', consisting of policies without restrictions on schools or social gatherings ($u^S = 1$, $u^O = 1$) and different levels $u^W$ of restrictions on workplace gatherings.
— 'Restricted work' policies, under which only 'essential' workers are allowed on-site work ($u^W = 0.2$), with either

 (i) no school restrictions ($u^S = 1$) and different levels of restrictions on social gatherings ($0.2 \leq u^O \leq 1$); or
 (ii) restrictions on social gatherings ($u^O = 0.3$, that is 'no pubs') and different levels of social distancing in school ($0 \leq u^S \leq 1$).

As figure 16d illustrates, 'school and pubs' and 'restricted work' policies are not efficient without shielding.

In the absence of shielding, social gatherings seem to be the main vector for contagion. When shielding measures are put in place, the social contacts associated with the elderly are reduced to the same level as under lockdown; in this case, contacts at work become the main vector of contagion.

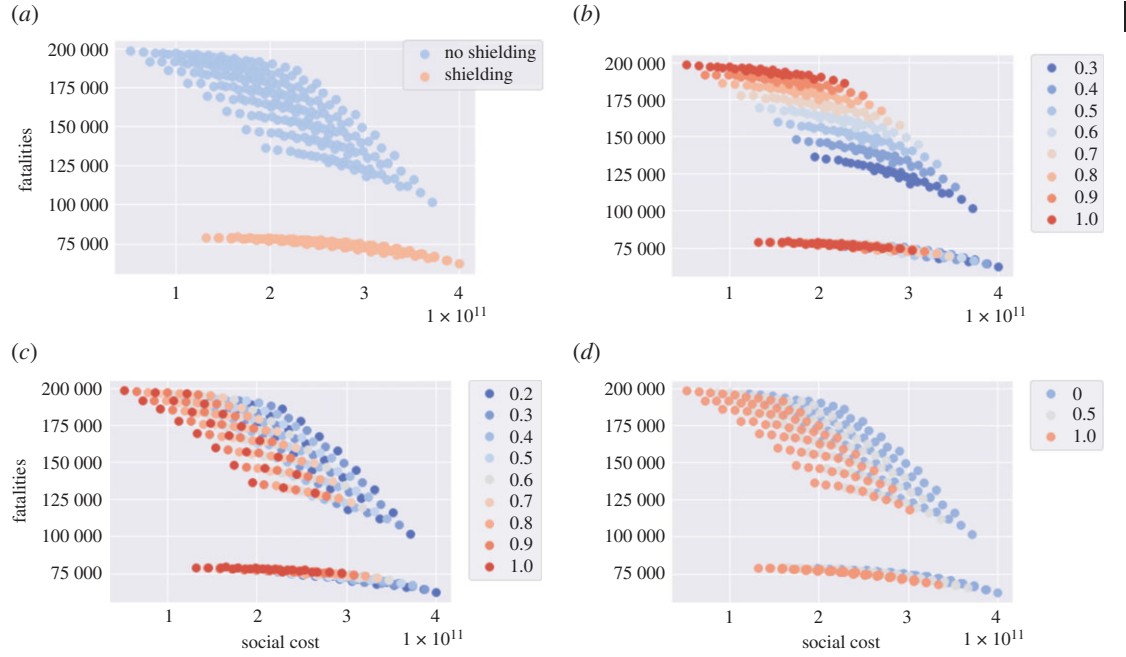

**Figure 13.** Efficiency plot of social cost against projected fatalities for the shielding measure and various values of $u^S$, $u^W$ and $u^O$ ($u^H = 1$ and $T = 105$). (a) Impact of the shielding measure for senior citizens, (b) social distancing outside work and school: impact of the parameter $u^O$, (c) social distancing at work: impact of the parameter $u^W$, (d) social distancing at school: impact of the parameter $u^S$.

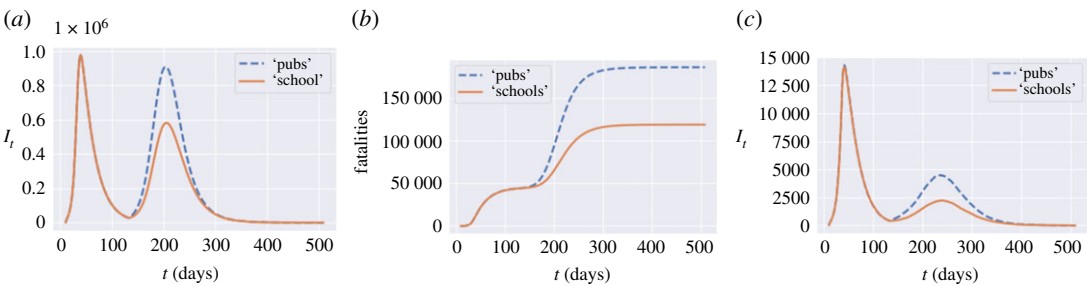

**Figure 14.** 'Open pubs' versus 'open schools' policy. (a) Infections in England, (b) fatalities in England, (c) infections: Oxfordshire (UKJ14).

## 5.3. Impact of parameter uncertainty

### 5.3.1. Uncertainty on the symptomatic ratio

The above results are sensitive to the value of the symptomatic ratios which, as noted in §3, are highly uncertain (table 4). Figure 17 shows the policy outcomes for low versus high symptomatic ratios across different compliance levels and lockdown duration. As observed in this figure, while the overall pattern of the efficiency diagram is similar, the projected fatality levels shift considerably depending on the assumption on the symptomatic ratio: from 50 000 to 200 000 for low symptomatic ratios to 126 000–430 000 for high symptomatic ratios.

### 5.3.2. Heterogeneity in infection rates

We now examine the impact of introducing different infection rates for symptomatic and asymptomatic carriers, as discussed in §3.4.

For the sake of brevity, we only show some sample results to illustrate the impact of heterogeneous infection rates. Figure 18 shows the comparison of the 'open schools' versus 'open pubs' policies described in §5.2.5, when infection rates are different for symptomatic and asymptomatic carriers. Comparison with figure 18 reveals a reduction of around 10% in fatalities but the overall picture

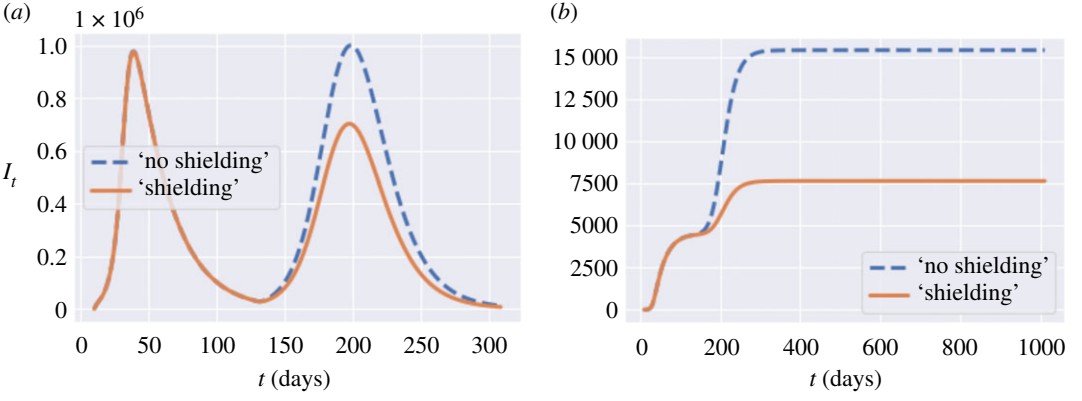

**Figure 15.** Comparison of policies with and without shielding in place, $u = (1, 0.0, 1.0, 0.5)$. Blue: no shielding; orange: shielding in place. (*a*) Symptomatic infections in England, (*b*) fatalities in England.

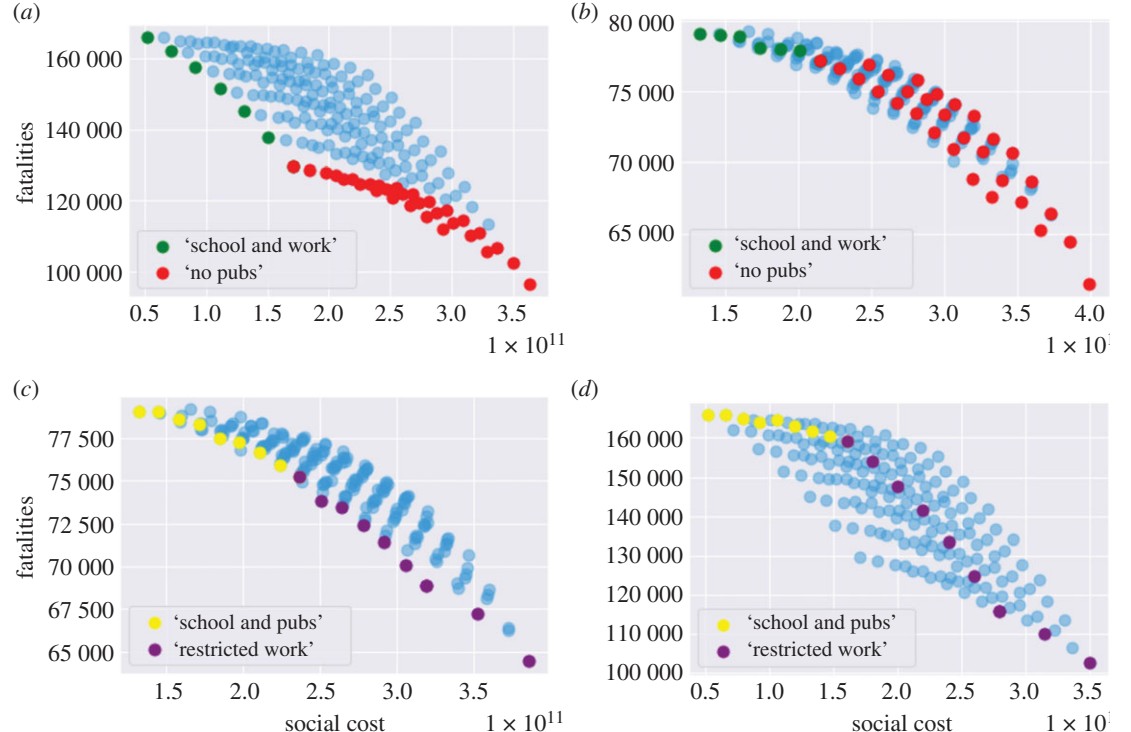

**Figure 16.** Impact of shielding on the efficiency frontier. (*a*) Efficient policies without shielding: 'school and work' and 'no pubs', (*b*) policies 'school and work' and 'no pubs' are not efficient when shielding is applied, (*c*) efficient policies with shielding: 'school and work' and 'no work', (*d*) policies 'school and pubs' and 'no work' are not efficient when shielding is removed.

remains similar: school closures are seen to be less effective than restrictions on non-work/school gatherings, as observed in §5.2.5.

Figure 19 shows the impact of heterogeneous infection rates on the efficiency diagram. As the figure illustrates, the overall pattern remains similar to the case of homogeneous infection rate, but the number of fatalities is reduced.

# 6. Adaptive mitigation policies

We now consider *adaptive* mitigation policies, in which the daily number of (national or regional) reported cases is used as a trigger for social distancing measures. Such policies have been recently implemented, in the UK and elsewhere, at a local or national level using a regional 'tier' system. We analyse the simplest version of such a tier system, namely a two-tier approach where a region is moved to 'Tier 2' when the number of cases goes above a threshold.

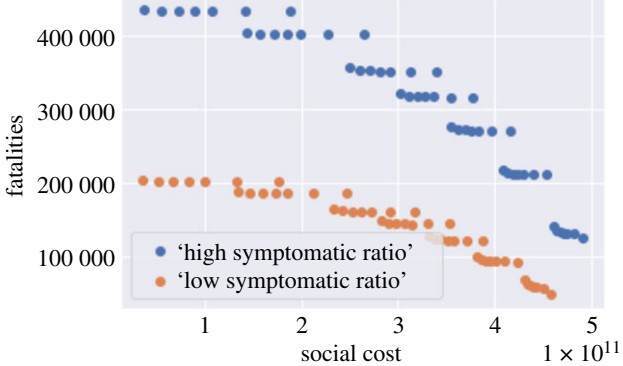

**Figure 17.** Trade-off between fatalities and social cost for a $T$-day lockdown followed by social distancing ($0.2 \leq m \leq 1$, $105 \leq T \leq 335$): low symptomatic ratio (orange) and high symptomatic ratio (blue).

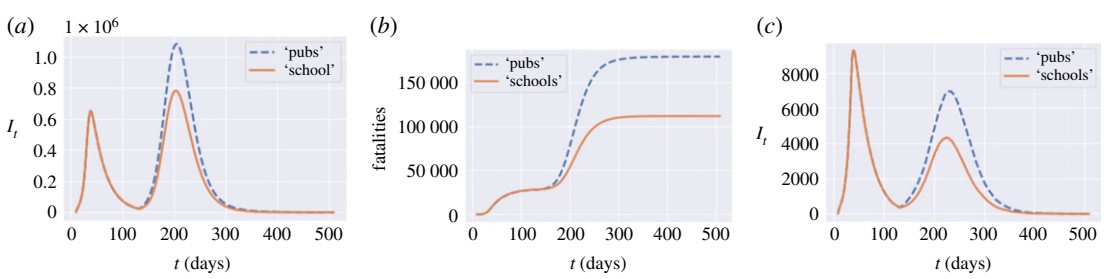

**Figure 18.** 'Open pubs' versus 'open schools' policy with different infection rates for symptomatic and asymptomatic population. (*a*) Infections in England, (*b*) fatalities in England, (*c*) infections: Oxfordshire (UKJ14).

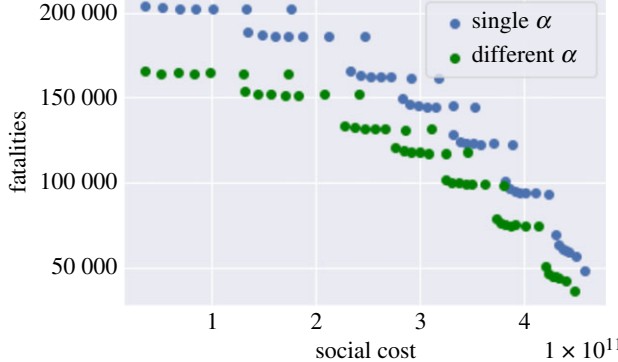

**Figure 19.** Trade-off between fatalities and social cost for a $T$-day lockdown followed by social distancing ($0.2 \leq m \leq 1$, $105 \leq T \leq 335$) under low symptomatic ratio: heterogeneous infection rates $\alpha_1 > \alpha_0$ for symptomatic/asymptomatic populations (green) and homogeneous infection rate $\alpha$ (blue).

We distinguish *centralized* policies, based on monitoring of national case numbers, from *decentralized* policies where monitoring and implementation of measures are done at the level of (NUTS-3) regions.

## 6.1. Centralized policies

We first consider centralized policies which monitor the number of daily reported cases at country level. Given a reporting probability $\pi(t)$ (see §4), given a number $r_t$ of new reported cases, the *estimated* number of cases is $r(t)/\pi(t)$. Whenever, the number of daily estimated cases (per 100 000 inhabitants) exceeds a threshold $B_{on}$, confinement measures are imposed for a minimum of $L$ days, until the number of daily (estimated) cases falls below the threshold $B_{off} < B_{on}$. Outside these lockdown periods, we assume social distancing is in place with a compliance level $m$; we use a default value of $m = 0.5$.

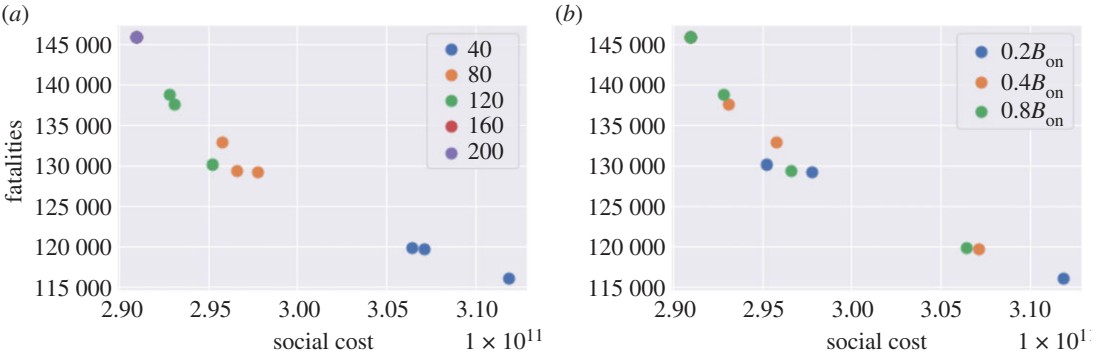

**Figure 20.** Social cost against fatalities when $m = 0.5$. (a) Influence of the threshold $B_{on}$ to resume lockdown, (b) influence of the threshold $B_{off}$ to lift lockdown.

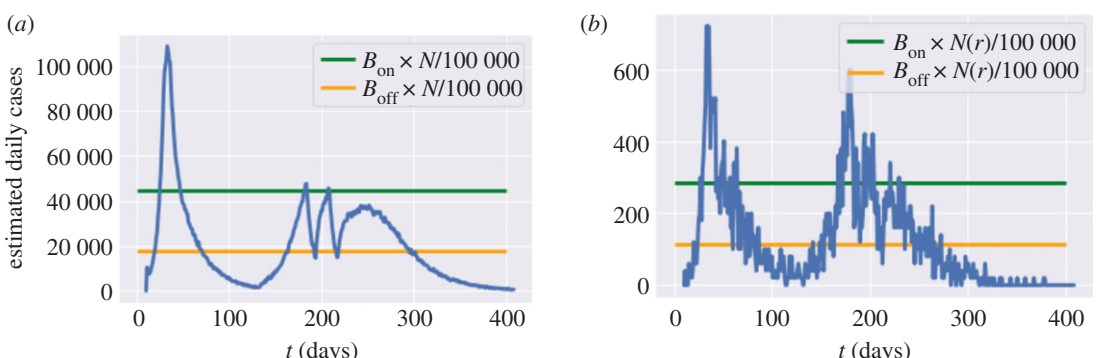

**Figure 21.** Simulation of reported cases in England and Leicester under a centralized triggering policy with $B_{on} = 80$, $B_{off} = 0.4 \times B_{on}$, $m = 0.5$ and no shielding. (a) Daily reported cases in England, (b) daily reported cases in Leicester.

This policy is implemented after the initial lockdown (that is, after 4 July 2020). In terms of the social contact matrix, we have, for $t > t_0 + T$,

$$
\begin{cases}
\sigma^r(t) = ((1 - m)l_r + md_r)\sigma, & \text{and } i_t = 0, & \text{if } \left(\frac{r(t)}{\pi(t)} \leq \frac{N}{100\,000}B_{off} \text{ and } \Pi_{s=t-L}^{t-1}i_s = 1\right) \\
& & \text{or } \left(\frac{r(t)}{\pi(t)} \leq \frac{N}{100\,000}B_{on} \text{ and } i_{t-1} = 0\right); \\
\sigma^r(t) = l_r \times \sigma, & \text{and } i_t = 1, & \text{if } \frac{r(t)}{\pi(t)} > \frac{N}{100\,000}B_{on} \text{ or } \Pi_{s=t-L}^{t-1}i_s = 1.
\end{cases}
\tag{6.1}
$$

Here, $T = 105$, $i_t$ is the indicator of whether lockdown is applied on day $t$, and $r(t)$ is the daily reported cases in England on day $t$. $\Pi_{s=t-L}^{t-1}i_s = 1$ if lockdown has been applied for $L$ consecutive days during the period $[t - L, t - 1]$.

We simulate the dynamics with various choices of $B_{off}$ and $B_{on}$:

— $B_{on} \in \{40, 80, 120, 160, 200\}$ (daily new cases per 100 000 inhabitants); and
— $B_{off} = 0.2\, B_{on}$, $B_{off} = 0.4\, B_{on}$ or $B_{off} = 0.8\, B_{on}$.

We assume that once a lockdown is triggered it lasts a minimum of $L = 7$ days and that, once lockdown is removed, individuals continue to observe social distancing as measured by the parameter $m \in [0, 1]$. Data on real-time mobility monitoring in the UK,[5] indicate mobility to be at 50% of normal level during the post-lockdown period, and thus we use $m = 0.5$ as a default value (figure 20).

### 6.1.1. Example

Figure 21 shows an example of such an adaptive policy, where lockdown is triggered when estimated daily cases exceeds 2240 nationally, and maintained until the count of new daily cases drops to 896.

[5]See https://www.oxford-covid-19.com/.

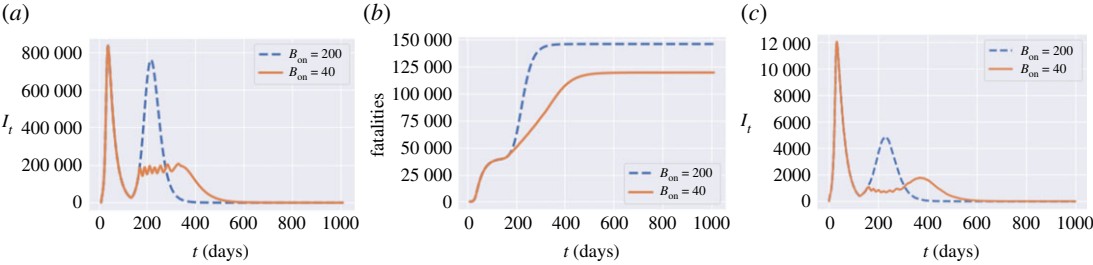

**Figure 22.** Comparison between triggering thresholds $B_{on} = 200$ and $B_{on} = 40$. (a) Dynamics of $I_t$ in England, (b) cumulative fatalities in England, (c) $I_t$ in Oxfordshire (UKJ14).

In the scenario shown in figure 21a, this results in two short lockdowns, totalling 19 days in all, which bring under control the national progression of the epidemic and avoid a 'second peak' at national level. However, as shown in figure 21b, this policy is less successful at regional level, resulting in a regional outbreak in Leicester.

### 6.1.2. Impact of the triggering threshold $B_{on}$

The trigger threshold $B_{on}$ has a significant impact on the efficiency of the policy. Smaller $B_{on}$ values correspond to more frequent lockdowns, leading to a larger social cost and fewer fatalities. Here, we compare the impact of the triggering threshold $B_{on}$ when $m = 0.5$ and $B_{off} = 0.4 \times B_{on}$ (figure 22).

We observe in our simulations a second peak in $I_t$ for England when $B_{on} = 200$, while we observe no second peak when $B_{on} = 40$. When $B_{on} = 40$, $I_t$ remains at level $2 \times 10^5$ with frequent interventions for 200 days and then decreases to zero. The social costs for policy $B_{on} = 200$ and policy $B_{on} = 40$ are 2.9 and 3.1, respectively. Policy $B_{on} = 40$ has 18% fewer fatalities compared to policy $B_{on} = 200$. Oxfordshire exhibits the same profile as England when $B_{on} = 10$. However, the shape of $I_t$ is different for $B_{on} = 40$ where Oxfordshire experiences a small outbreak around day 350.

In summary, smaller $B_{on}$ values correspond to more frequent lockdowns and result in damping or elimination of the 'second peak'.

### 6.1.3. Impact of demographic granularity

Several studies on the impact of public health policies on COVID-19 dynamics have used less granular models with fewer age groups [10]. To assess whether such coarse-graining may result in a loss of accuracy for the model projections, we have compared our present model, which has 16 age groups, with coarse-grained versions of the model in which all individuals in the 20–59 age range are grouped into two age groups (leading to a total of five age groups) or a single group (leading to 4 age groups).[6] Parameters for the coarse-grained models are obtained as population-weighted averages of the granular model.

Comparison of model projections, shown in figure 23, indicate that the results are robust to changes in model granularity. Some quantitative differences may emerge when assessing the impact of targeted policies, but the overall dynamics of infections, cases and fatalities are rather insensitive to the demographic granularity.

## 6.2. Decentralized policies: regional tier system

We now consider a decentralized version of the above policies, based on monitoring of regional number of cases as triggers for regional confinement measures. In terms of the social contact matrices, we have, for $t > t_0 + T$,

$$
\begin{cases}
\sigma^k(t) = ((1-m)l_k + md_k)\sigma, & i_t^k = 0, & \text{if } \left( \frac{r_t(k)}{\pi(t)} \leq \frac{N(k)}{100\,000} B_{off} \quad \text{and} \quad \Pi_{s=t-L}^{t-1} i_s^k = 1 \right) \\
& & \text{or } \left( \frac{r_t(k)}{\pi(t)} \leq \frac{N(k)}{100\,000} B_{on} \quad \text{and} \quad i_{t-1}^k = 0 \right); \\
\sigma^k(t) = l_k \times \sigma, & \text{and } i_t^k = 1, & \text{if } \frac{r_t(k)}{\pi(t)} > \frac{N(k)}{100\,000} B_{on} \quad \text{or} \quad \Pi_{s=t-L}^{t-1} i_s^k = 1.
\end{cases}
$$

---

[6]This model was implemented in a previous version of this paper: https://www.medrxiv.org/content/10.1101/2020.08.26.20182477v2

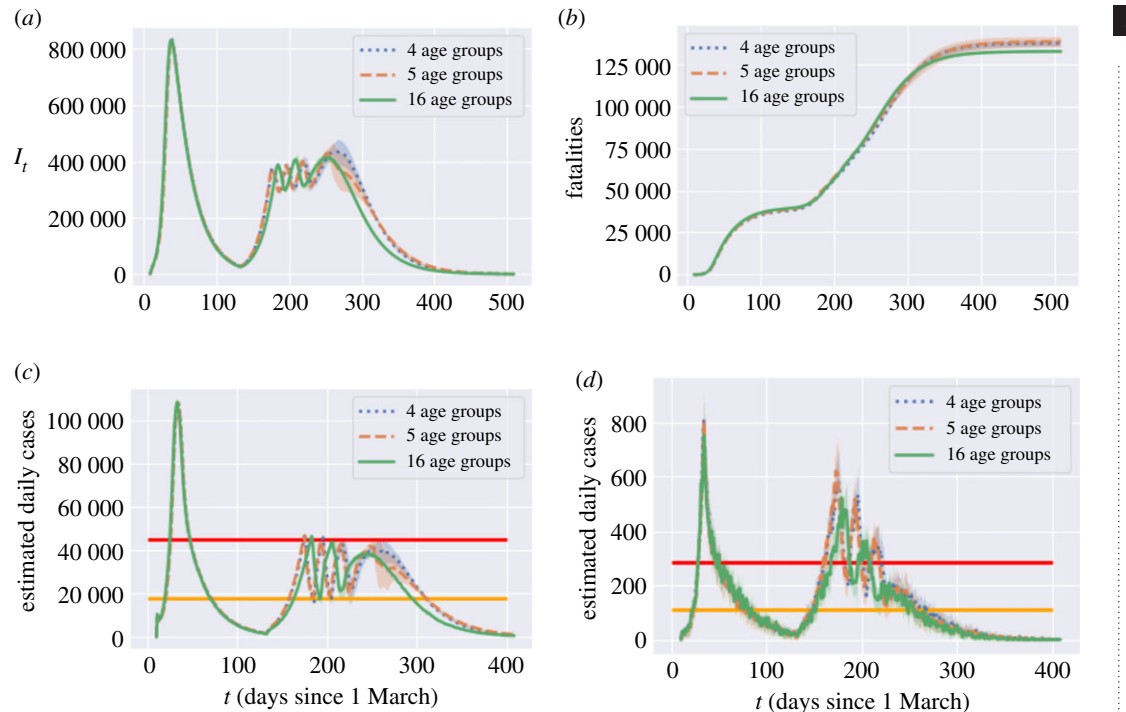

**Figure 23.** Impact of model granularity: projections for an adaptive policy with $B_{on} = 80$, $B_{off} = 0.4 \times B_{on}$, $m = 0.5$ and no shielding. (a) Symptomatic infections ($I_t$) in England, (b) projected fatalities, (c) projections for reported cases: England, (c) projections for reported cases: Leicester.

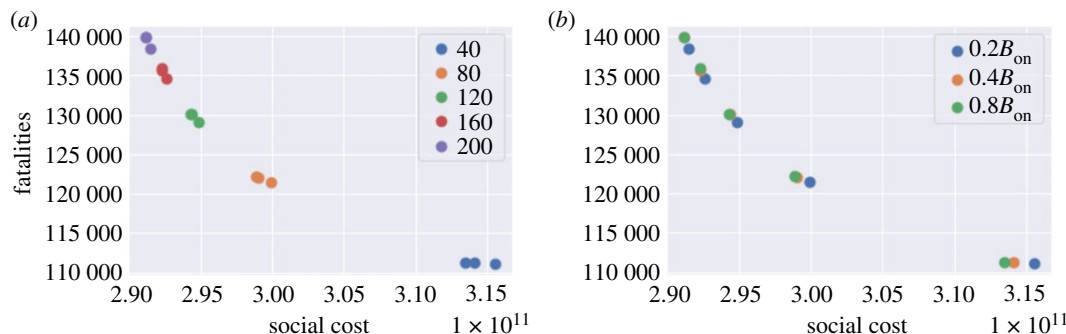

**Figure 24.** Decentralized confinement triggered by regional daily case numbers: social cost versus fatalities ($m = 0.5$). (a) Influence of the threshold $B_{on}$ for triggering lockdown, (b) Influence of the threshold $B_{off}$ for lifting lockdown.

Here, $i_t^k$ is the indicator of whether lockdown is applied in region $k$ on day $t$ and $r_t(k)$ is the daily number of cases reported in region $k$ on day $t$. The term $\Pi_{s=t-L}^{t-1} i_s^k$ is used to track if lockdown has been applied in region $k$ for $L$ consecutive days during $[t-L, t-1]$. We use the same values of $B_{on}$ and $B_{off}$ as in §6.1 (see figure 24).

Figure 25 compares the outcomes of centralized and decentralized triggering policies. Decentralized policies are observed to always improve over centralized policies.

As an example, for $B_{on} = 80$ and $B_{off} = 0.4 B_{on}$ fatalities in England are 133 000 under the centralized policy and 122 000 under the decentralized policy, that is 8% lower.

Figure 26 compares regional fatalities per 100 000 habitants for these policies. For more than 90% of the regions, decentralized measures lead to fewer fatalities. The most effective reductions are in Dorset, South West England (UKK22) with 23% fewer fatalities and in Cornwall and Isles of Scilly (UKK30) with 21% fewer fatalities. There are a few exceptions (see regions in light blue in figure 26c). These regions are already under control before adaptive policies are applied. Therefore, the improvement of moving from centralized policy to decentralized policy is limited.

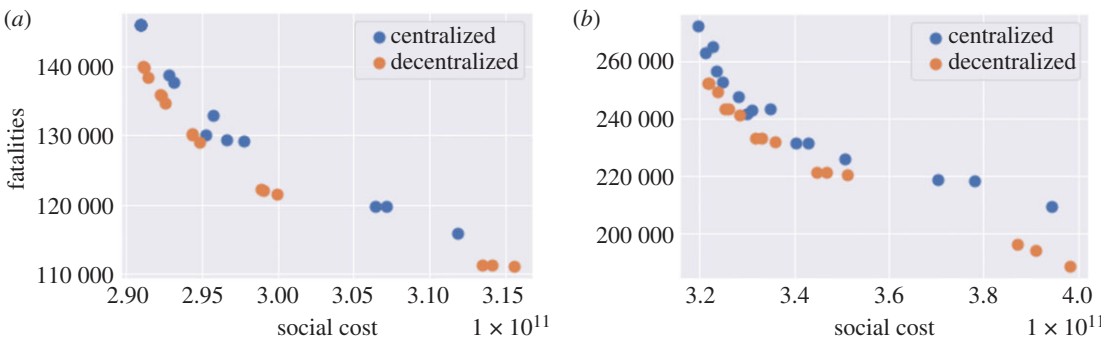

**Figure 25.** Efficiency analysis for centralized (blue) and decentralized (orange) adaptive mitigation policies. Outcomes are averaged across 100 simulated scenarios. (*a*) Low symptomatic ratios, (*b*) high symptomatic ratios.

**Figure 26.** Fatalities per 100 000 inhabitants for centralized (*a*) versus regional (*b*) adaptive mitigation policies. Same triggering thresholds are used in both cases: $B_{on} = 80$ and $B_{off} = 0.4\,B_{on}$. (*a*) Centralized (country-level) adaptive policy, (*b*) decentralized (regional) adaptive policy, (*c*) increase in fatalities (per 100 000 inhabitants) when moving from regional to centralized policy.

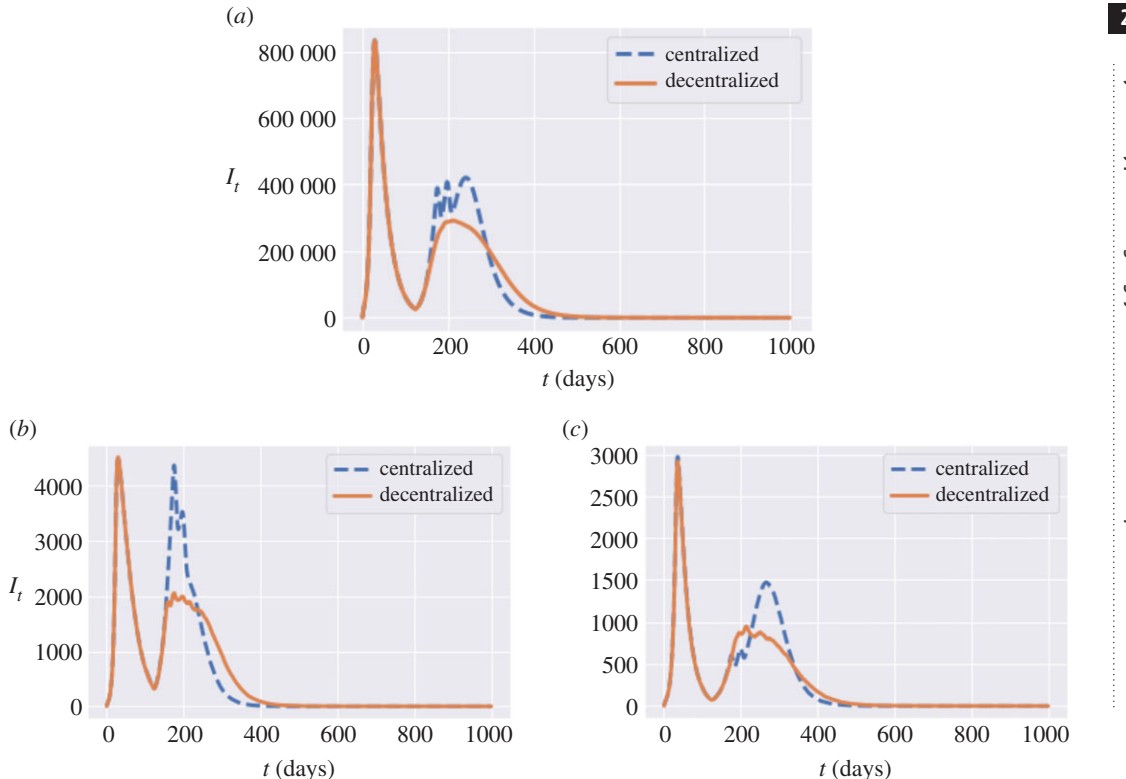

**Figure 27.** Number of infected individuals under centralized (blue dashed line) and decentralized (orange solid line) policies. Same triggering thresholds are used in both cases: $B_{on} = 80$ and $B_{off} = 0.4\ B_{on}$. (a) Number of symptomatic individuals ($I_t$) in England under centralized and decentralized policies, (b) number of symptomatic individuals ($I_t$) in Leicester (UKF21), (c) number of symptomatic individuals ($I_t$) in York (UKE21).

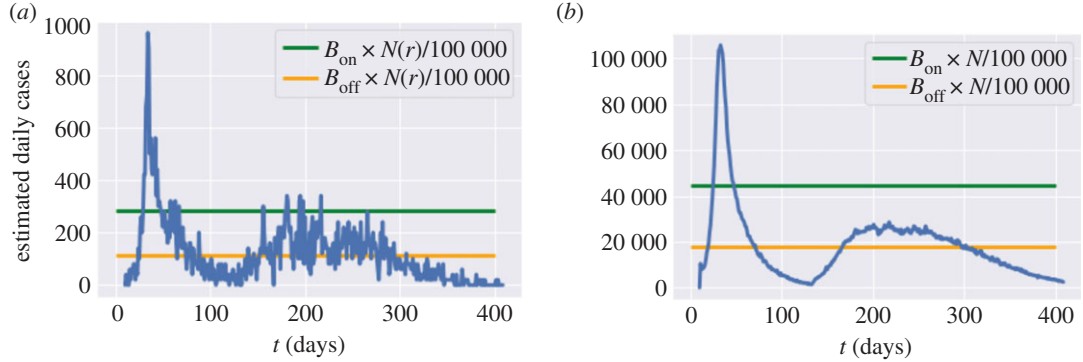

**Figure 28.** Reported cases in England and Leicester under a decentralized triggering policy: average of 50 simulated scenarios with $B_{on} = 80$, $B_{off} = 0.4 \times B_{on}$, $m = 0.5$, no shielding. (a) Daily reported cases in Leicester, (b) daily reported cases in England.

Figure 27a compares the dynamics of symptomatic infections ($I_t$) for the same example. There is a reduction of 100 000 in the amplitude of the second peak value when moving from the centralized policy to decentralized one. Decentralized policy also damps the second-peak values in most of the regions. Similar effects are observed for York (figure 27c) and Leicester (figure 27b).

On 29 June 2020, Leicester became the first city in Britain to be placed in a local lockdown, after public health officials voiced concern at the city's alarming rise in COVID-19 cases. Earlier in June, the Government announced that parts of the city would be released from lockdown, while a 'targeted' approach will see pockets remain under tighter restrictions. Our simulations indicate a 60% reduction of the second-peak value in Leicester when a decentralized policy is implemented (figure 27b).

**Table 10.** Summary of outcomes for different policies, starting from the same initial conditions on 4 July 2020.

| policy | $I_t$ (1 August) | $A_t$ (1 August) | fatalities (1 August) | max $I_t$ (2nd peak) | social cost ($10^{11}$) | projected fatalities (1000 days) |
|---|---|---|---|---|---|---|
| confinement followed by strict social distancing ($m = 0.3$) | 47 400 | 188 700 | 39 400 | 255 700 | 3.8 | 96 600 |
| confinement followed by moderate social distancing ($m = 0.5$) | 98 400 | 392 400 | 40 700 | 766 800 | 2.9 | 146 100 |
| pre-planned | 84 700 | 360 100 | 45 500 | 613 300 | 2.9 | 122 900 |
| centralized triggering | 80 300 | 321 200 | 40 500 | 423 200 | 3.0 | 133 500 |
| decentralized triggering | 80 100 | 320 200 | 40 400 | 292 200 | 3.0 | 122 100 |
| decentralized triggering and shielding | 55 000 | 266 100 | 39 700 | 267 700 | 3.4 | 65 900 |
| 'protect lives' | 25 900 | 118 400 | 39 600 | 63 900 | 4.3 | 51 700 |

### 6.2.1. Example

Figure 28 shows an example of such a decentralized triggering policy, with the same triggering thresholds as in the centralized example in figure 21. At regional level, we see in figure 28a that this policy is more successful than the centralized policy in taming the local outbreaks in Leicester, substantially reducing the second peak through 4 one-week regional lockdowns. At the national level, this results in a strong damping of 'second wave' infections, as shown in figure 28b (compare with figure 21a).

## 6.3. Adaptive versus pre-planned policies

Figure 29 compares the health outcome and social cost of the efficient policies considered in §§5.2, 6.1 and 6.2. The efficient frontier of pre-planned policies are among policies with $u^S \in \{0, 0.5, 1\}$, $0.2 \leq u^W \leq 1.0$ and $0.3 \leq u^O \leq 1.0$. For centralized and decentralized policies, $m = 0.25, 0.5, 0.75, 1$; $B_{on} = 80, 160, 120, 160, 200$; and $B_{off} = p \times B_{on}$ with $p = 0.2, 0.4, 0.8$.

We observe that
— adaptive policies, in which measures are triggered when the number of daily new cases exceeds a threshold, are more efficient than pre-planned policies; and
— as shown in figure 29a,b, a decentralized policy is more efficient than both centralized policy and pre-planned policy.

In table 10, we provide a summary of outcomes for five different types of policies;
— confinement of $T = 105$ days followed by social distancing ($m = 0.3$ or $m = 0.5$), no shielding;
— pre-planned policy: social distancing at work and school ($u^H = 1$, $u^S = 0.5$, $u^W = 0.5$), restrictions on social gatherings ($u^O = 0.3$) and no shielding.
— centralized and decentralized triggering policies (§§6.1 and 6.2) with $m = 0.5$, $B_{on} = 80$, $B_{off} = 0.4 B_{on}$ and no shielding;
— decentralized triggering combined with shielding of elderly populations: $m = 0.5$, $B_{on} = 80$, $B_{off} = 0.4 B_{on}$;
— 'protect lives' policy: in the range of efficient policies, the one which results in the fewest fatalities is a decentralized triggering policy with $B_{on} = 40$, $B_{off} = 0.2 B_{on}$ (so more frequent triggering of confinement measures than the above), high degree of social distancing ($m = 0.25$) and shielding of elderly populations. This policy corresponds to the point in the lower right corner of figure 29b. The social cost is 4.52, which is much higher than for the other considered policies.

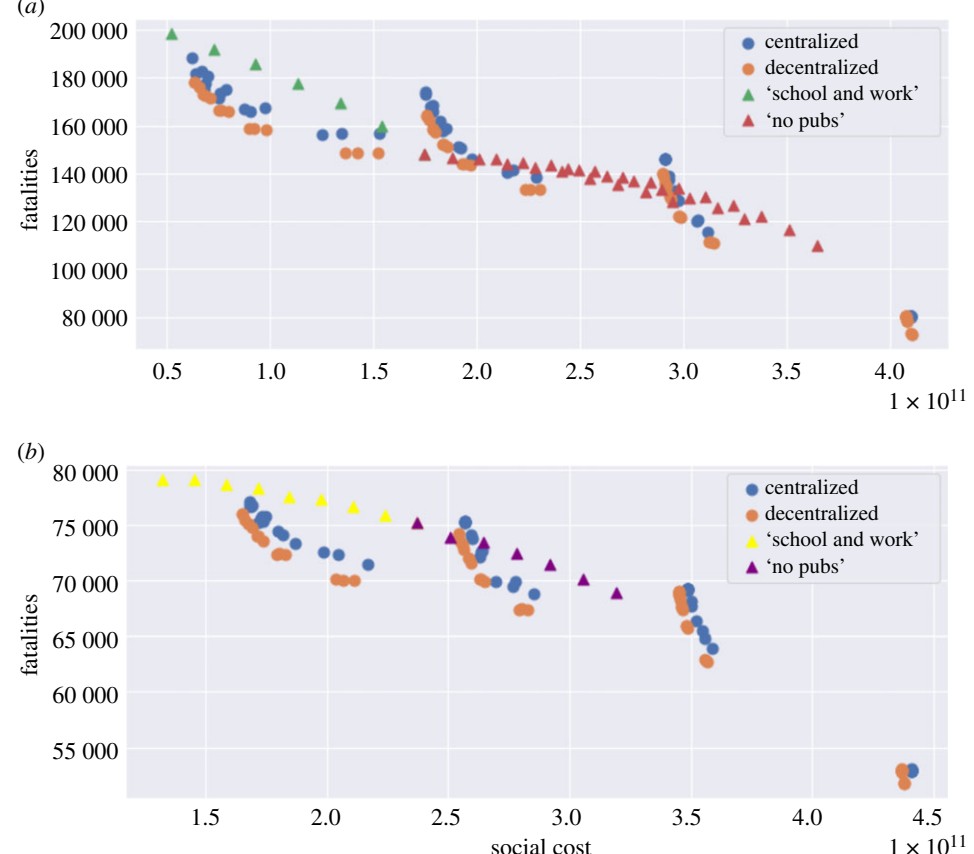

**Figure 29.** Efficiency plot: pre-planned versus adaptive mitigation policies. (*a*) No Shielding, (*b*) shielding.

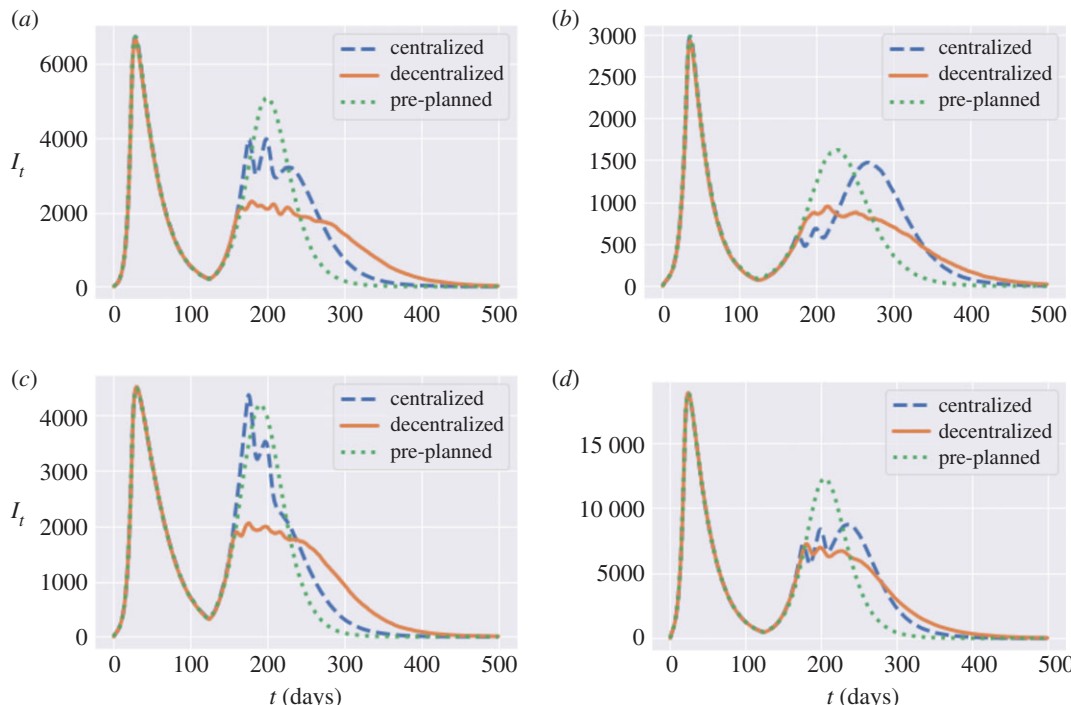

**Figure 30.** Regional comparison of pre-planned and adaptive mitigation policies. (*a*) $I_t$ in Mid Lancashire, (*b*) $I_t$ in York, (*c*) $I_t$ in Leicester, (*d*) $I_t$ in Birmingham.

Outcomes are averaged across 50 scenarios, starting from the same initial conditions on 4 July (end of the UK lockdown).

### 6.3.1. Regional outcomes

Comparing the regional outcomes of the centralized, decentralized and pre-planned policies displayed in table 10 shows that the decentralized triggering policies are able in many cases to considerably damp the 'second wave' of infections. Figure 30 illustrates this in the case of Mid Lancashire, York, Leicester and Birmingham: the decentralized triggering policy reduces the second peak amplitude by around one half compared to the pre-planned policy.

Data accessibility. All data used in this study are publicly available and the data sources are explicitly mentioned in the manuscript. Data and relevant code for this research work are stored in GitHub: https://github.com/RenyuanXu/COVID-19 and have been archived within the Zenodo repository: https://zenodo.org/record/4292569 (doi:10.5281/zenodo.4292569).
Authors' contributions. All authors contributed equally to the analysis and writing of the manuscript.
Competing interests. We declare we have no competing interests.
Funding. No funding has been received for this article.

# Appendix A. Demographic regions

Table 11 details the used mapping between Upper Tier Local Authority (UTLA) region codes and the Nomenclature of Territorial Units for Statistics at level 3 (NUTS-3) codes.[7] If more than one UTLA region falls within the boundary of a single NUTS-3 region, the data is then aggregated. On the other hand, if a single UTLA region lies within more than one NUTS-3 region, the data are distributed among NUTS-3 regions in proportion to the total number of people living in each region.

**Table 11.** Mapping between the Upper Tier Local Authority (UTLA) regions and the Nomenclature of Territorial Units for Statistics at level 3 codes (NUTS-3).

| UTLA Code | UTLA region name | NUTS-3 code mapping |
|---|---|---|
| E06000001 | Hartlepool | UKC11 |
| E06000002 | Middlesbrough | UKC12 |
| E06000003 | Redcar and Cleveland | UKC12 |
| E06000004 | Stockton-on-Tees | UKC11 |
| E06000005 | Darlington | UKC13 |
| E06000006 | Halton | UKD71 |
| E06000007 | Warrington | UKD61 |
| E06000008 | Blackburn with Darwen | UKD41 |
| E06000009 | Blackpool | UKD42 |
| E06000010 | Kingston upon Hull, City of | UKE11 |
| E06000011 | East Riding of Yorkshire | UKE12 |
| E06000012 | North East Lincolnshire | UKE13 |
| E06000013 | North Lincolnshire | UKE13 |
| E06000014 | York | UKE21 |
| E06000015 | Derby | UKF11 |
| E06000016 | Leicester | UKF21 |
| E06000017 | Rutland | UKF22 |

(Continued.)

---

[7]See https://geoportal.statistics.gov.uk/datasets/c893dfece45f465f857ac34641041863_0 for a lookup table used in the process of mapping.

| UTLA Code | UTLA region name | NUTS-3 code mapping |
|---|---|---|
| E06000018 | Nottingham | UKF14 |
| E06000019 | Herefordshire, County of | UKG11 |
| E06000020 | Telford and Wrekin | UKG21 |
| E06000021 | Stoke-on-Trent | UKG23 |
| E06000022 | Bath and North East Somerset | UKK12 |
| E06000023 | Bristol, City of | UKK11 |
| E06000024 | North Somerset | UKK12 |
| E06000025 | South Gloucestershire | UKK12 |
| E06000026 | Plymouth | UKK41 |
| E06000027 | Torbay | UKK42 |
| E06000030 | Swindon | UKK14 |
| E06000031 | Peterborough | UKH11 |
| E06000032 | Luton | UKH21 |
| E06000033 | Southend-on-Sea | UKH31 |
| E06000034 | Thurrock | UKH32 |
| E06000035 | Medway | UKJ41 |
| E06000036 | Bracknell Forest | UKJ11 |
| E06000037 | West Berkshire | UKJ11 |
| E06000038 | Reading | UKJ11 |
| E06000039 | Slough | UKJ11 |
| E06000040 | Windsor and Maidenhead | UKJ11 |
| E06000041 | Wokingham | UKJ11 |
| E06000042 | Milton Keynes | UKJ12 |
| E06000043 | Brighton and Hove | UKJ21 |
| E06000044 | Portsmouth | UKJ31 |
| E06000045 | Southampton | UKJ32 |
| E06000046 | Isle of Wight | UKJ34 |
| E06000047 | County Durham | UKC14 |
| E06000049 | Cheshire East | UKD62 |
| E06000050 | Cheshire West and Chester | UKD63 |
| E06000051 | Shropshire | UKG22 |
| E06000052 | Cornwall and Isles of Scilly | UKK30 |
| E06000054 | Wiltshire | UKK15 |
| E06000055 | Bedford | UKH24 |
| E06000056 | Central Bedfordshire | UKH25 |
| E06000057 | Northumberland | UKC21 |
| E06000058 | Bournemouth and Poole | UKK21 |
| E06000059 | Dorset | UKK22 |
| E08000001 | Bolton | UKD36 |
| E08000002 | Bury | UKD37 |
| E08000003 | Manchester | UKD33 |
| E08000004 | Oldham | UKD37 |
| E08000005 | Rochdale | UKD37 |

(*Continued.*)

| UTLA Code | UTLA region name | NUTS-3 code mapping |
| --- | --- | --- |
| E08000006 | Salford | UKD34 |
| E08000007 | Stockport | UKD35 |
| E08000008 | Tameside | UKD35 |
| E08000009 | Trafford | UKD34 |
| E08000010 | Wigan | UKD36 |
| E08000011 | Knowsley | UKD71 |
| E08000012 | Liverpool | UKD72 |
| E08000013 | St. Helens | UKD71 |
| E08000014 | Sefton | UKD73 |
| E08000015 | Wirral | UKD74 |
| E08000016 | Barnsley | UKE31 |
| E08000017 | Doncaster | UKE31 |
| E08000018 | Rotherham | UKE31 |
| E08000019 | Sheffield | UKE32 |
| E08000021 | Newcastle upon Tyne | UKC22 |
| E08000022 | North Tyneside | UKC22 |
| E08000023 | South Tyneside | UKC22 |
| E08000024 | Sunderland | UKC23 |
| E08000025 | Birmingham | UKG31 |
| E08000026 | Coventry | UKG33 |
| E08000027 | Dudley | UKG36 |
| E08000028 | Sandwell | UKG37 |
| E08000029 | Solihull | UKG32 |
| E08000030 | Walsall | UKG38 |
| E08000031 | Wolverhampton | UKG39 |
| E08000032 | Bradford | UKE41 |
| E08000033 | Calderdale | UKE44 |
| E08000034 | Kirklees | UKE44 |
| E08000035 | Leeds | UKE42 |
| E08000036 | Wakefield | UKE45 |
| E08000037 | Gateshead | UKC22 |
| E09000001 | City of London | UKI31 |
| E09000002 | Barking and Dagenham | UKI52 |
| E09000003 | Barnet | UKI71 |
| E09000004 | Bexley | UKI51 |
| E09000005 | Brent | UKI72 |
| E09000006 | Bromley | UKI61 |
| E09000007 | Camden | UKI31 |
| E09000008 | Croydon | UKI62 |
| E09000009 | Ealing | UKI73 |
| E09000010 | Enfield | UKI54 |
| E09000011 | Greenwich | UKI51 |
| E09000012 | Hackney | UKI41 |

| UTLA Code | UTLA region name | NUTS-3 code mapping |
|---|---|---|
| E09000013 | Hammersmith and Fulham | UKI33 |
| E09000014 | Haringey | UKI43 |
| E09000015 | Harrow | UKI74 |
| E09000016 | Havering | UKI52 |
| E09000017 | Hillingdon | UKI74 |
| E09000018 | Hounslow | UKI75 |
| E09000019 | Islington | UKI43 |
| E09000020 | Kensington and Chelsea | UKI33 |
| E09000021 | Kingston upon Thames | UKI63 |
| E09000022 | Lambeth | UKI45 |
| E09000023 | Lewisham | UKI44 |
| E09000024 | Merton | UKI63 |
| E09000025 | Newham | UKI41 |
| E09000026 | Redbridge | UKI53 |
| E09000027 | Richmond upon Thames | UKI75 |
| E09000028 | Southwark | UKI44 |
| E09000029 | Sutton | UKI63 |
| E09000030 | Tower Hamlets | UKI42 |
| E09000031 | Waltham Forest | UKI53 |
| E09000032 | Wandsworth | UKI34 |
| E09000033 | Westminster | UKI32 |
| E10000002 | Buckinghamshire | UKJ13 |
| E10000003 | Cambridgeshire | UKH12 |
| E10000006 | Cumbria | UKD11, UKD12 |
| E10000007 | Derbyshire | UKF13, UKF12 |
| E10000008 | Devon | UKK43 |
| E10000011 | East Sussex | UKJ22 |
| E10000012 | Essex | UKH37, UKH34, UKH35, UKH36 |
| E10000013 | Gloucestershire | UKK13, UKK12 |
| E10000014 | Hampshire | UKJ36, UKJ37, UKJ35 |
| E10000015 | Hertfordshire | UKH23 |
| E10000016 | Kent | UKJ43, UKJ44, UKJ45, UKJ46 |
| E10000017 | Lancashire | UKD45, UKD46, UKD47, UKD44 |
| E10000018 | Leicestershire | UKF22 |
| E10000019 | Lincolnshire | UKE13, UKF30 |
| E10000020 | Norfolk | UKH15, UKH17, UKH16 |
| E10000021 | Northamptonshire | UKF24, UKF25 |
| E10000023 | North Yorkshire | UKE22 |
| E10000024 | Nottinghamshire | UKF15, UKF16 |
| E10000025 | Oxfordshire | UKJ14 |
| E10000027 | Somerset | UKK12, UKK23 |
| E10000028 | Staffordshire | UKG24 |
| E10000029 | Suffolk | UKH14 |

(*Continued.*)

| UTLA Code | UTLA region name | NUTS-3 code mapping |
|---|---|---|
| E10000030 | Surrey | UKJ25, UKJ26 |
| E10000031 | Warwickshire | UKG13 |
| E10000032 | West Sussex | UKJ28, UKJ27 |
| E10000034 | Worcestershire | UKG12 |

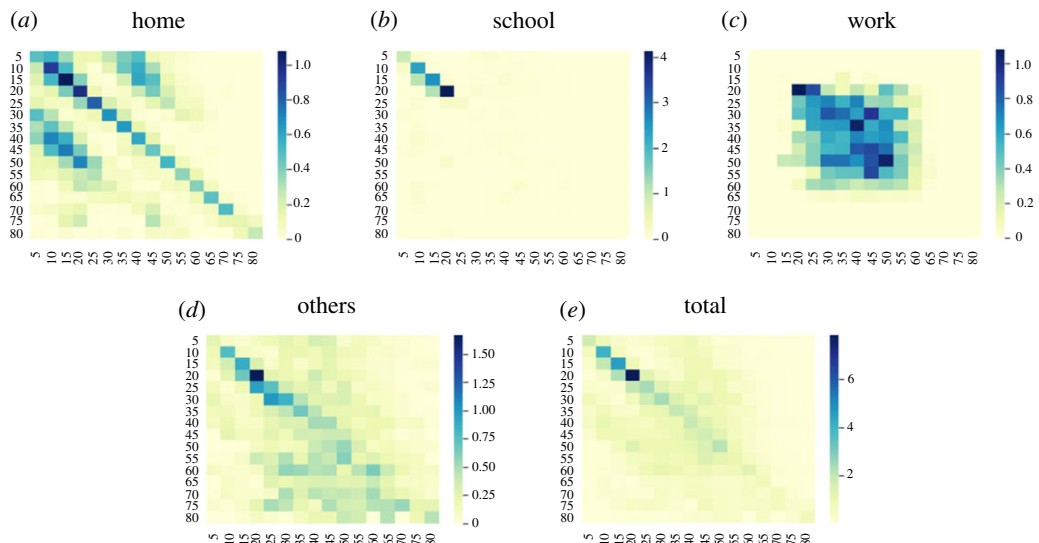

**Figure 31.** Baseline social contact matrices with 16 age groups. (*a*) Home, (*b*) school, (*c*) work, (*d*) other, (*e*) total.

# Appendix B. Baseline parameters for social contact rates

This appendix outlines the sources used for the baseline social contact rate parameters. In particular, two sources have been used: the POLYMOD study [41], processed using the methodology of PyRoss [15], and the BBC Pandemic study [56] is used as a robustness check. It should be noted that these parameters are used as a baseline, and a further detailed calibration is carried out region by region to account for heterogeneity of social contact patterns across UK regions. We use estimates for social contact rates across the 16 age groups (detailed in table 2) given in Mossong *et al.* [41]:

$$
\sigma = \begin{bmatrix}
1.92 & 0.81 & 0.47 & 0.30 & 0.49 & 0.79 & 0.89 & 1.07 & 0.44 & 0.27 & 0.35 & 0.27 & 0.22 & 0.15 & 0.10 & 0.02 \\
0.78 & 6.64 & 1.24 & 0.58 & 0.49 & 0.72 & 1.09 & 1.40 & 1.10 & 0.36 & 0.35 & 0.23 & 0.35 & 0.24 & 0.07 & 0.23 \\
0.42 & 1.16 & 6.85 & 1.30 & 0.25 & 0.37 & 0.57 & 1.10 & 1.18 & 0.64 & 0.35 & 0.35 & 0.2 & 0.2 & 0.17 & 0.14 \\
0.26 & 0.52 & 1.26 & 6.71 & 1.24 & 0.72 & 0.47 & 0.87 & 0.97 & 0.97 & 0.52 & 0.31 & 0.2 & 0.26 & 0.24 & 0.28 \\
0.43 & 0.44 & 0.24 & 1.26 & 2.59 & 1.36 & 0.84 & 0.76 & 0.83 & 0.93 & 0.63 & 0.5 & 0.31 & 0.22 & 0.16 & 0.17 \\
0.73 & 0.68 & 0.38 & 0.76 & 1.42 & 1.83 & 1.13 & 0.92 & 0.9 & 0.92 & 0.85 & 0.72 & 0.45 & 0.38 & 0.18 & 0.12 \\
0.73 & 0.93 & 0.53 & 0.44 & 0.79 & 1.02 & 1.67 & 1.27 & 0.98 & 0.72 & 0.7 & 0.63 & 0.48 & 0.27 & 0.09 & 0.27 \\
0.79 & 1.06 & 0.89 & 0.74 & 0.63 & 0.74 & 1.12 & 1.5 & 1.27 & 0.86 & 0.63 & 0.55 & 0.53 & 0.43 & 0.14 & 0.31 \\
0.32 & 0.83 & 0.97 & 0.83 & 0.69 & 0.73 & 0.87 & 1.28 & 1.35 & 1.21 & 0.7 & 0.55 & 0.55 & 0.35 & 0.33 & 0.43 \\
0.24 & 0.32 & 0.62 & 0.96 & 0.91 & 0.87 & 0.75 & 1.02 & 1.41 & 1.87 & 0.75 & 0.64 & 0.51 & 0.26 & 0.32 & 0.33 \\
0.34 & 0.34 & 0.37 & 0.57 & 0.68 & 0.88 & 0.81 & 0.82 & 0.90 & 0.82 & 0.74 & 0.98 & 0.46 & 0.31 & 0.28 & 0.76 \\
0.24 & 0.20 & 0.35 & 0.32 & 0.50 & 0.69 & 0.67 & 0.66 & 0.66 & 0.66 & 0.91 & 1.17 & 0.73 & 0.43 & 0.20 & 0.46 \\
0.24 & 0.40 & 0.25 & 0.25 & 0.38 & 0.54 & 0.65 & 0.80 & 0.82 & 0.65 & 0.53 & 0.91 & 0.65 & 0.55 & 0.30 & 0.66 \\
0.19 & 0.31 & 0.29 & 0.39 & 0.32 & 0.52 & 0.42 & 0.74 & 0.60 & 0.39 & 0.42 & 0.63 & 0.64 & 0.70 & 0.52 & 0.20 \\
0.15 & 0.11 & 0.28 & 0.40 & 0.26 & 0.29 & 0.17 & 0.28 & 0.66 & 0.55 & 0.44 & 0.34 & 0.40 & 0.60 & 0.59 & 0.57 \\
0.01 & 0.18 & 0.12 & 0.24 & 0.14 & 0.10 & 0.24 & 0.32 & 0.43 & 0.28 & 0.60 & 0.39 & 0.45 & 0.12 & 0.29 & 0.86
\end{bmatrix}
$$

We use the Bayesian hierarchical framework provided by the PyRoss library [40] to decompose contact rates into 'work', 'home', 'school' and 'other' [57]. The results are provided below, and also visualized in figure 31.

$$
\sigma^H =
\begin{bmatrix}
0.48 & 0.55 & 0.33 & 0.13 & 0.14 & 0.28 & 0.41 & 0.49 & 0.11 & 0.07 & 0.04 & 0.02 & 0.01 & 0.00 & 0.00 & 0.00 \\
0.26 & 0.92 & 0.52 & 0.12 & 0.03 & 0.17 & 0.45 & 0.58 & 0.32 & 0.07 & 0.03 & 0.01 & 0.00 & 0.00 & 0.00 & 0.00 \\
0.17 & 0.54 & 1.08 & 0.39 & 0.04 & 0.01 & 0.22 & 0.59 & 0.49 & 0.13 & 0.04 & 0.02 & 0.00 & 0.01 & 0.00 & 0.00 \\
0.09 & 0.15 & 0.42 & 0.98 & 0.13 & 0.03 & 0.06 & 0.25 & 0.42 & 0.21 & 0.07 & 0.05 & 0.01 & 0.02 & 0.00 & 0.00 \\
0.17 & 0.08 & 0.07 & 0.36 & 0.80 & 0.21 & 0.07 & 0.04 & 0.16 & 0.28 & 0.08 & 0.09 & 0.02 & 0.01 & 0.00 & 0.00 \\
0.49 & 0.30 & 0.04 & 0.07 & 0.13 & 0.66 & 0.21 & 0.03 & 0.01 & 0.05 & 0.17 & 0.11 & 0.03 & 0.00 & 0.00 & 0.00 \\
0.32 & 0.47 & 0.27 & 0.08 & 0.05 & 0.09 & 0.64 & 0.15 & 0.03 & 0.00 & 0.01 & 0.04 & 0.02 & 0.00 & 0.00 & 0.00 \\
0.38 & 0.70 & 0.56 & 0.20 & 0.02 & 0.01 & 0.09 & 0.59 & 0.15 & 0.00 & 0.02 & 0.01 & 0.01 & 0.01 & 0.00 & 0.00 \\
0.17 & 0.52 & 0.73 & 0.42 & 0.07 & 0.00 & 0.09 & 0.19 & 0.46 & 0.10 & 0.02 & 0.00 & 0.03 & 0.01 & 0.01 & 0.00 \\
0.13 & 0.15 & 0.33 & 0.71 & 0.35 & 0.08 & 0.01 & 0.08 & 0.07 & 0.53 & 0.11 & 0.03 & 0.01 & 0.00 & 0.00 & 0.01 \\
0.12 & 0.10 & 0.18 & 0.26 & 0.31 & 0.07 & 0.07 & 0.04 & 0.08 & 0.06 & 0.37 & 0.13 & 0.01 & 0.00 & 0.00 & 0.00 \\
0.02 & 0.01 & 0.11 & 0.28 & 0.23 & 0.18 & 0.06 & 0.02 & 0.01 & 0.08 & 0.09 & 0.39 & 0.10 & 0.00 & 0.00 & 0.00 \\
0.02 & 0.00 & 0.05 & 0.03 & 0.09 & 0.10 & 0.11 & 0.07 & 0.08 & 0.02 & 0.04 & 0.11 & 0.48 & 0.06 & 0.01 & 0.00 \\
0.05 & 0.07 & 0.11 & 0.15 & 0.02 & 0.01 & 0.02 & 0.10 & 0.22 & 0.03 & 0.02 & 0.05 & 0.08 & 0.51 & 0.05 & 0.00 \\
0.00 & 0.02 & 0.16 & 0.25 & 0.01 & 0.00 & 0.00 & 0.00 & 0.30 & 0.08 & 0.03 & 0.00 & 0.06 & 0.14 & 0.16 & 0.09 \\
0.02 & 0.00 & 0.04 & 0.00 & 0.02 & 0.00 & 0.04 & 0.00 & 0.07 & 0.05 & 0.08 & 0.00 & 0.00 & 0.00 & 0.11 & 0.27
\end{bmatrix}
$$

$$
\sigma^S =
\begin{bmatrix}
0.97 & 0.15 & 0.01 & 0.03 & 0.01 & 0.09 & 0.13 & 0.09 & 0.04 & 0.03 & 0.00 & 0.00 & 0.01 & 0.00 & 0.00 & 0.00 \\
0.24 & 2.35 & 0.06 & 0.00 & 0.02 & 0.04 & 0.06 & 0.06 & 0.06 & 0.05 & 0.04 & 0.00 & 0.00 & 0.00 & 0.00 & 0.00 \\
0.00 & 1.12 & 2.56 & 0.12 & 0.01 & 0.05 & 0.04 & 0.12 & 0.09 & 0.07 & 0.03 & 0.02 & 0.01 & 0.00 & 0.00 & 0.00 \\
0.04 & 0.08 & 1.17 & 4.14 & 0.06 & 0.12 & 0.08 & 0.08 & 0.07 & 0.05 & 0.04 & 0.03 & 0.00 & 0.00 & 0.00 & 0.00 \\
0.00 & 0.13 & 0.00 & 0.27 & 0.23 & 0.01 & 0.02 & 0.03 & 0.01 & 0.01 & 0.01 & 0.00 & 0.00 & 0.00 & 0.00 & 0.00 \\
0.11 & 0.07 & 0.01 & 0.00 & 0.00 & 0.06 & 0.04 & 0.06 & 0.04 & 0.03 & 0.00 & 0.00 & 0.00 & 0.00 & 0.00 & 0.00 \\
0.00 & 0.10 & 0.01 & 0.01 & 0.01 & 0.00 & 0.04 & 0.00 & 0.02 & 0.00 & 0.00 & 0.00 & 0.00 & 0.00 & 0.00 & 0.00 \\
0.04 & 0.17 & 0.07 & 0.03 & 0.00 & 0.02 & 0.05 & 0.07 & 0.07 & 0.05 & 0.00 & 0.01 & 0.00 & 0.01 & 0.00 & 0.00 \\
0.07 & 0.10 & 0.03 & 0.00 & 0.04 & 0.06 & 0.03 & 0.06 & 0.02 & 0.01 & 0.01 & 0.01 & 0.00 & 0.00 & 0.00 & 0.00 \\
0.02 & 0.00 & 0.02 & 0.21 & 0.00 & 0.00 & 0.05 & 0.01 & 0.07 & 0.05 & 0.06 & 0.04 & 0.01 & 0.00 & 0.00 & 0.00 \\
0.00 & 0.00 & 0.00 & 0.00 & 0.00 & 0.00 & 0.00 & 0.00 & 0.02 & 0.00 & 0.00 & 0.04 & 0.00 & 0.00 & 0.00 & 0.00 \\
0.05 & 0.14 & 0.05 & 0.00 & 0.00 & 0.00 & 0.03 & 0.00 & 0.15 & 0.03 & 0.00 & 0.08 & 0.00 & 0.00 & 0.00 & 0.00 \\
0.00 & 0.13 & 0.00 & 0.00 & 0.02 & 0.01 & 0.05 & 0.11 & 0.02 & 0.03 & 0.01 & 0.01 & 0.00 & 0.00 & 0.00 & 0.00 \\
0.00 & 0.06 & 0.04 & 0.00 & 0.00 & 0.00 & 0.04 & 0.03 & 0.00 & 0.00 & 0.00 & 0.03 & 0.00 & 0.00 & 0.00 & 0.00 \\
0.00 & 0.00 & 0.00 & 0.00 & 0.00 & 0.00 & 0.00 & 0.00 & 0.00 & 0.00 & 0.00 & 0.00 & 0.00 & 0.00 & 0.00 & 0.00 \\
0.00 & 0.00 & 0.00 & 0.06 & 0.00 & 0.00 & 0.00 & 0.00 & 0.00 & 0.00 & 0.00 & 0.00 & 0.00 & 0.00 & 0.00 & 0.00
\end{bmatrix}
$$

$$
\sigma^W =
\begin{bmatrix}
0.00 & 0.00 & 0.00 & 0.00 & 0.00 & 0.00 & 0.00 & 0.00 & 0.00 & 0.00 & 0.00 & 0.00 & 0.00 & 0.00 & 0.00 & 0.00 \\
0.00 & 0.00 & 0.00 & 0.00 & 0.00 & 0.00 & 0.00 & 0.00 & 0.00 & 0.00 & 0.00 & 0.00 & 0.00 & 0.00 & 0.00 & 0.00 \\
0.00 & 0.00 & 0.02 & 0.00 & 0.00 & 0.00 & 0.10 & 0.00 & 0.04 & 0.00 & 0.00 & 0.00 & 0.00 & 0.00 & 0.00 & 0.00 \\
0.00 & 0.00 & 0.01 & 1.08 & 0.87 & 0.28 & 0.11 & 0.28 & 0.15 & 0.48 & 0.32 & 0.06 & 0.02 & 0.00 & 0.00 & 0.00 \\
0.00 & 0.00 & 0.02 & 0.44 & 0.64 & 0.71 & 0.56 & 0.71 & 0.35 & 0.37 & 0.17 & 0.16 & 0.02 & 0.00 & 0.00 & 0.00 \\
0.00 & 0.00 & 0.01 & 0.49 & 0.61 & 0.84 & 0.77 & 0.62 & 0.95 & 0.54 & 0.55 & 0.09 & 0.02 & 0.00 & 0.00 & 0.00 \\
0.00 & 0.00 & 0.05 & 0.10 & 0.59 & 0.63 & 0.65 & 1.05 & 0.58 & 0.69 & 0.27 & 0.21 & 0.02 & 0.00 & 0.00 & 0.00 \\
0.00 & 0.00 & 0.02 & 0.33 & 0.37 & 0.59 & 0.54 & 0.70 & 0.62 & 0.66 & 0.52 & 0.19 & 0.01 & 0.00 & 0.00 & 0.00 \\
0.00 & 0.00 & 0.01 & 0.23 & 0.39 & 0.53 & 0.55 & 0.83 & 0.88 & 0.77 & 0.35 & 0.25 & 0.02 & 0.00 & 0.00 & 0.00 \\
0.00 & 0.00 & 0.26 & 0.28 & 0.39 & 0.76 & 0.76 & 0.68 & 0.85 & 1.04 & 0.42 & 0.19 & 0.03 & 0.00 & 0.00 & 0.00 \\
0.00 & 0.00 & 0.00 & 0.00 & 0.12 & 0.43 & 0.47 & 0.48 & 0.88 & 0.52 & 0.42 & 0.16 & 0.05 & 0.00 & 0.00 & 0.00 \\
0.00 & 0.00 & 0.01 & 0.09 & 0.33 & 0.36 & 0.31 & 0.25 & 0.29 & 0.33 & 0.30 & 0.14 & 0.01 & 0.00 & 0.00 & 0.00 \\
0.00 & 0.00 & 0.00 & 0.00 & 0.03 & 0.07 & 0.03 & 0.09 & 0.07 & 0.05 & 0.06 & 0.04 & 0.00 & 0.00 & 0.00 & 0.00 \\
0.00 & 0.00 & 0.00 & 0.00 & 0.00 & 0.00 & 0.00 & 0.00 & 0.00 & 0.00 & 0.00 & 0.00 & 0.00 & 0.00 & 0.00 & 0.00 \\
0.00 & 0.00 & 0.00 & 0.00 & 0.00 & 0.00 & 0.00 & 0.00 & 0.00 & 0.00 & 0.00 & 0.00 & 0.00 & 0.00 & 0.00 & 0.00 \\
0.00 & 0.00 & 0.00 & 0.00 & 0.00 & 0.00 & 0.00 & 0.00 & 0.00 & 0.00 & 0.00 & 0.00 & 0.00 & 0.00 & 0.00 & 0.00
\end{bmatrix}
$$

$$\sigma^O = \begin{bmatrix}
0.26 & 0.10 & 0.05 & 0.13 & 0.19 & 0.26 & 0.19 & 0.34 & 0.31 & 0.07 & 0.15 & 0.11 & 0.06 & 0.04 & 0.04 & 0.01 \\
0.18 & 0.77 & 0.13 & 0.09 & 0.06 & 0.22 & 0.23 & 0.18 & 0.26 & 0.17 & 0.16 & 0.06 & 0.09 & 0.06 & 0.04 & 0.01 \\
0.15 & 0.35 & 0.88 & 0.31 & 0.09 & 0.21 & 0.09 & 0.17 & 0.26 & 0.13 & 0.11 & 0.05 & 0.06 & 0.02 & 0.04 & 0.05 \\
0.04 & 0.26 & 0.68 & 1.67 & 0.27 & 0.20 & 0.16 & 0.37 & 0.25 & 0.21 & 0.16 & 0.08 & 0.06 & 0.04 & 0.01 & 0.00 \\
0.15 & 0.04 & 0.18 & 0.96 & 0.75 & 0.37 & 0.24 & 0.30 & 0.16 & 0.32 & 0.09 & 0.14 & 0.08 & 0.03 & 0.02 & 0.05 \\
0.25 & 0.09 & 0.08 & 0.20 & 0.99 & 0.85 & 0.46 & 0.27 & 0.29 & 0.32 & 0.20 & 0.10 & 0.09 & 0.05 & 0.02 & 0.00 \\
0.16 & 0.17 & 0.09 & 0.18 & 0.32 & 0.44 & 0.67 & 0.44 & 0.23 & 0.23 & 0.22 & 0.25 & 0.10 & 0.06 & 0.01 & 0.02 \\
0.13 & 0.22 & 0.10 & 0.10 & 0.24 & 0.30 & 0.31 & 0.52 & 0.51 & 0.25 & 0.15 & 0.20 & 0.21 & 0.10 & 0.11 & 0.04 \\
0.00 & 0.25 & 0.15 & 0.15 & 0.27 & 0.18 & 0.36 & 0.35 & 0.39 & 0.39 & 0.24 & 0.20 & 0.10 & 0.02 & 0.05 & 0.00 \\
0.00 & 0.01 & 0.03 & 0.13 & 0.19 & 0.13 & 0.17 & 0.39 & 0.46 & 0.56 & 0.32 & 0.12 & 0.16 & 0.10 & 0.06 & 0.06 \\
0.03 & 0.03 & 0.10 & 0.39 & 0.30 & 0.48 & 0.24 & 0.46 & 0.41 & 0.60 & 0.23 & 0.29 & 0.20 & 0.14 & 0.12 & 0.01 \\
0.11 & 0.06 & 0.06 & 0.11 & 0.36 & 0.57 & 0.55 & 0.46 & 0.49 & 0.22 & 0.46 & 0.64 & 0.40 & 0.20 & 0.12 & 0.11 \\
0.00 & 0.09 & 0.07 & 0.03 & 0.29 & 0.22 & 0.28 & 0.31 & 0.32 & 0.27 & 0.28 & 0.40 & 0.26 & 0.09 & 0.11 & 0.08 \\
0.02 & 0.11 & 0.06 & 0.06 & 0.11 & 0.49 & 0.29 & 0.46 & 0.42 & 0.43 & 0.39 & 0.52 & 0.29 & 0.22 & 0.14 & 0.06 \\
0.07 & 0.03 & 0.09 & 0.24 & 0.51 & 0.44 & 0.24 & 0.14 & 0.49 & 0.36 & 0.34 & 0.47 & 0.48 & 0.55 & 0.24 & 0.25 \\
0.00 & 0.00 & 0.04 & 0.11 & 0.07 & 0.04 & 0.13 & 0.22 & 0.20 & 0.41 & 0.35 & 0.00 & 0.47 & 0.00 & 0.19 & 0.47
\end{bmatrix}$$

Note that these baseline values are modulated to reflect regional differences, using the approach described in §3.

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
