## [Peer Review File · Royal Society Open Science]

Review History

RSOS-201535.R0 (Original submission)

Review form: Reviewer 1

Is the manuscript scientifically sound in its present form?

Yes

Are the interpretations and conclusions justified by the results?

No

Is the language acceptable?

Yes

Do you have any ethical concerns with this paper?

No

Have you any concerns about statistical analyses in this paper?

Yes

Recommendation?

Major revision is needed (please make suggestions in comments)

Comments to the Author(s)

This paper presents a SIR-like model of COVID dynamics in the UK that incorporates social structure and geographic heterogeneity. Overall, the model is well put together and the consideration the authors give to spatial heterogeneity is a valuable contribution. The authors use this model to make a medley of different points including: (1) estimates of real world values (such as the total number of people infected), (2) counterfactual estimates (how many lives the lock-down saved), (3) theoretical consideration (how many days it is possible to observe 0 cases while the epidemic is ongoing) (4) heterogeneity in model parameters across regions and (5) effects of different intervention strategies. Of these 5 points by (5) is by far the most compelling. The authors do a thorough analysis of mitigation strategies and also quantify the social cost which has seen little quantitative discussion in related work.

As it stands the manuscript is very disorganized. Points 1-4 should be either shortened/removed or considerably strengthened and/or turned into a separate manuscripts.

With regards to specific analyses (1-4) my criticism in brief is:

- (1) Estimates of real world values should be subjected to extensive robustness analysis and predictive (forecasting) validation. I believe this is beyond the scope of this paper.
- (2) Counterfactual estimates are generally difficult to support. Since the epidemic is not over we cannot purport to know how many lives were saved by the lockdown since it depends the future epidemic trajectory.
- (3) The discussion about the possible number of days with 0 infections is unlikely to be real world relevant (see discussion below) and doesn't leverage the UK specific model. If the authors chose to pursue this it is better to spin this off into another paper that includes theoretical analysis similar to (<https://personalpages.manchester.ac.uk/staff/thomas.house/blog/why-zero-is-so-hard.html>)
- (4) The spatial heterogeneity claims need to be strengthened with statistical analysis

Major comments about the model itself:

1: The authors need a very clear table of all the parameters of the model and whether these are fixed or estimated (an extended Table 3) and this should come much earlier in the paper.

2: I don't understand why the contact matrix from Mossong et al was binned into coarser age groups than the 5-year groups in original data. This seems entirely unnecessary for the purpose of this model yet incurs the cost of considerably under-estimating heterogeneity and thus overestimating epidemic size. I am in particular concerned about the binning of 20-60 year olds into one group. The number of contacts vary by several fold within that age range. The authors should re-run their analysis with the original bins. I expect that many of the key results regarding mitigation will be strengthened.

3: Much of the results and the discussion is devoted to asymptomatic infections. The authors state at the beginning: "The probability p that an infected individual develops symptoms is an important parameter for epidemic dynamics." It is not clear to me why this is universally the case without some qualifying assumption. These assumptions are never stated.

The importance of asymptomatic cases has to be attributed to:

- Asymptomatic individuals are less infectious
- Asymptomatic individuals recover more quickly

-Asymptomatic infections are less detectable.

If none of these hold than modeling asymptomatic infections is irrelevant. The authors should qualify which of these are being modeled, and where the alternative parameters come from. The overall infectiousness and recovery rates can be easily sources from literature but how were the alternative asymptomatic parameters determined. These should be added with citations to the master parameter table.

Comments regarding the analysis

1: Overall I think that analyses claiming to produce accurate estimates of real world parameters need to show parameter/model robustness and also predictive value. At the very least, the authors should acknowledged that absolute numbers of infections or deaths computed by the model are not statements of fact. While I think it is reasonable to compare mitigation strategies within the context of this model saying that X number of people will die under scenario Y is not appropriate. These numbers are only as accurate as the underlying assumptions about model parameters which are uncertain. Furthermore there are many sources of unknown heterogeneity and this will in general reduce the real world epidemic size over what is predicted. The authors do not make an attempt to fit unmodeled heterogeneity (as is done in <https://www.medrxiv.org/content/10.1101/2020.04.27.20081893v3>) and I don't think this is necessary here but it does imply that statements about epidemic size and number of deaths cannot be taken as facts about the real world. Overall, I find this part of the analysis the least compelling and it distracts with the much more interesting (IMO) analysis of mitigation strategies at the end.

2: A related or separate point is that the following statement as it is misleading and undermines the otherwise high standard of scientific rigor in this paper.

"We estimate that, in absence of social distancing and confinement measures, the number of fatalities in England may have exceeded 216,000 by August 1, 2020, indicating that the lockdown has saved more than 174,000 lives." Besides the more global point (1) above it is not possible to say that X number of lives have been saved since this depends on the future number of deaths. Counterfactual claims are generally very difficult to support and should in my opinion be avoided altogether.

3: There are multiple issues with the figures throughout the paper.

-As a general rule the legend of the figure should be completely self sufficient in order to understand the content (What is A,H, and F in Figure 19?) The reader should not have to look at the text. Ideally, the authors would also give one or two sentences describing the point of the figure. For many figures in the paper I don't know what the reader should take away from it.

-Many of the figures are missing legend labels (perhaps a technical error with PDF generation???)

-There are too many figures. I suggest focusing on at most 7 highly compelling figures and making the rest supplemental.

-Some specific comments are below under "minor comments" but as there were a lot of omissions in the figures the list is likely not exhaustive and I suggest checking each one carefully.

4: The analysis presented in Figure 2 need to be fleshed out more. As far as I can tell the method does not return a posterior distribution. Can the authors determine if these values are indeed statistically different. It would be nice to back up these results with some external data. One of

the regions with the highest multiplier (small region NNW of London) doesn't seem to correspond to any known population center, how should we interpret this result?

5: The analysis presented in Figure 5 and Table 4 is not convincing. The authors purport to show some variation in the regional multiplier. Even if it could be statistically shown that these are indeed different (see comment about Figure 2) the interpretation is not straight forward. Certainly, it is premature to call this "compliance" as it is not clear if these values are determined by relative changes in behavior rather than baseline variation in behavior/social norms. For example, for communities with relatively small rates of recreational socializing the multiplicative effect of mitigations will also be smaller since there is some minimal amount of necessary socializing that must persist. Overall I found this section of the paper to be among the least compelling.

6: The discussion of how it is possible to have a long run of 0 cases while the epidemic is progressing (section 4.2) should be shortened or removed. It is obvious that this *can* happen for some parameters (such as very low detection rate) but it also doesn't appear to be relevant to the real world. The authors bring up S. Korea as an example but cases never went to 0 there. I am not aware of a location where community acquired cases went to 0 and then re-emerged from a community source. While there was some speculation about the New Zealand second wave genomic analysis strongly favors the re-introduction hypothesis.

7. Authors need to provide source code, input data and examples.

Minor comments

-In equation 4.2 the authors use what they refer to as $\bar{i}f$ -- The (average) infection fatality rate. However, this is strongly dependent on the age distribution of the infected individuals. How was this take into account?

-The analysis in Figure 9 seems to be considerably influenced by the 7 day periodicity (in the correlation is locally maximal when the periodicity is aligned, giving local peaks at 1,7, and 14). It would be better to correlate the window average.

-Figure 3a is not explained at all.

-In figure 3b the simulation and the data should span the same interval on the x-axis.

-What is the pink region in Figure 8

-What is on the X-axis in Figure 10.

-Figure 16 is not referenced in the text. It is unclear what it show. If these are data fits it would be better to show a per-region scatter plot against real data. The map display, while visually pleasing is not informative.

-I don't understand the purpose of Figure 17. Is there supposed to be a difference between (a) and (b) -- it is not visible by eye. Also legend labels are missing.

-I don't understand figure 18. There are no labels on the legend. I am assuming the orange line is the fit. Why does the first panel show results of multiple simulations but only an average is shown for the second panel. Why does the simulation not go for as long as the data?

-I don't see any purpose to figures 19-22 other than to show that the models are different. There is no systematic differences and the results are not compared to real data.

-small thing but in 2.6 I would change "man×day" to "person×day" since that most accurately reflects the correct unit.

-The authors assume that adaptive policies monitor case numbers. (As an aside I would avoid using R_t to denote this threshold to avoid confusion with the reproductive number). However, it is also possible to monitor hospitalizations that are less subject to reporting probability. This should be discussed.

Review form: Reviewer 2

Is the manuscript scientifically sound in its present form?

Yes

Are the interpretations and conclusions justified by the results?

Yes

Is the language acceptable?

Yes

Do you have any ethical concerns with this paper?

No

Have you any concerns about statistical analyses in this paper?

No

Recommendation?

Accept with minor revision (please list in comments)

Comments to the Author(s)

The paper presents an extended compartmental model for SARS-COV-2 in the UK stratified by both age and region. The model is then used to assess the effectiveness of different control measures in reducing disease impact. This metric is then considered against the social cost of each set of measures in order to determine a sense of efficiency.

I generally found the article to be well presented and insightful with sensible modelling assumptions throughout. I believe the assessment of control measures will be of interest to others and have just a few suggestions for improvement prior to publication.

The authors spend a long time in the early parts of the paper (sections 3-6) discussing model fitting and presenting the benefits of a metapopulation approach. While much of this is certainly useful and informative, such models are certainly not new and similar approaches have already been extensively applied to the Covid pandemic. What this paper adds to the discussion is in its assessment of control measures and I wonder whether it would be beneficial to move a lot of this early discussion to appendices or a supplement. The paper is not short and I think this would help give it a stronger message.

As far as the model itself is concerned, I thought the greatest deficiency for its use in the context of epidemic control, was the lack of any quarantining dynamic. Having symptomatic individuals (or even exposed households) reducing their contact may have a huge impact. Though I recognise no model is perfect, it may be worth acknowledging this for future inclusion.

In section 6.1 I was a little unclear on the length of the period of social distancing after lockdown. I assume no matter the level of compliance, if this period is long enough the social cost would be greater than that of the lockdown itself? Could this be clarified?

In section 7.1 there is a discussion of increasing the testing capacity. Is the effect of this identical to changing the triggering thresholds? Since the testing capacity increases over time then might there be some temporal effects on the triggering thresholds? This may be worth mentioning.

Finally a different colouring (over a larger spectrum) would be beneficial for the map figures, in their current form they appear rather homogenous.

Decision letter (RSOS-201535.R0)

Dear Professor Cont

The Editors assigned to your paper RSOS-201535 "Modelling COVID-19 contagion: risk assessment and targeted mitigation policies" have now received comments from reviewers and would like you to revise the paper in accordance with the reviewer comments and any comments from the Editors. Please note this decision does not guarantee eventual acceptance.

Please submit your revised manuscript and required files (see below) no later than 21 days from today's (ie 03-Nov-2020) date. Note: the ScholarOne system will 'lock' if submission of the revision is attempted 21 or more days after the deadline. If you do not think you will be able to meet this deadline please contact the editorial office immediately.

Kind regards,
Royal Society Open Science Editorial Office

on behalf of Professor Tim Rogers (Associate Editor) and Mark Chaplain (Subject Editor)
 openscience@royalsociety.org

Associate Editor: 2
 Comments to the Author:
 Dear Authors,

My opinion is that this manuscript meets the criteria to be put to out peer review at Open Science, however, I feel I need to give you a fair warning that I expect it to take a very long time. All of the experts in this field are, as I'm sure you are aware, enormously busy doing their own modelling, analysis and forecasting of the ongoing pandemic and government response. Recent past experience suggests that it will be very difficult to find qualified reviewers who are able to give their time to refereeing your manuscript.

I would suggest that if you want your work to influence policy making in the UK, then you need to also pursue other routes. For example, consider contacting members of the RAMP project:
<https://royalsociety.org/topics-policy/Health%20and%20wellbeing/ramp/>

Best wishes
 Tim Rogers

Reviewer comments to Author:
 Reviewer: 1

Comments to the Author(s)

This paper presents a SIR-like model of COVID dynamics in the UK that incorporates social structure and geographic heterogeneity. Overall, the model is well put together and the consideration the authors give to spatial heterogeneity is a valuable contribution. The authors use this model to make a medley of different points including: (1) estimates of real world values (such as the total number of people infected), (2) counterfactual estimates (how many lives the lock-down saved), (3) theoretical consideration (how many days it is possible to observe 0 cases while the epidemic is ongoing) (4) heterogeneity in model parameters across regions and (5) effects of different intervention strategies. Of these 5 points by (5) is by far the most compelling. The authors do a thorough analysis of mitigation strategies and also quantify the social cost which has seen little quantitative discussion in related work.

As it stands the manuscript is very disorganized. Points 1-4 should be either shortened/removed or considerably strengthened and/or turned into a separate manuscripts.

With regards to specific analyses (1-4) my criticism in brief is:

- (1) Estimates of real world values should be subjected to extensive robustness analysis and predictive (forecasting) validation. I believe this is beyond the scope of this paper.
- (2) Counterfactual estimates are generally difficult to support. Since the epidemic is not over we cannot purport to know how many lives were saved by the lockdown since it depends the future epidemic trajectory.
- (3) The discussion about the possible number of days with 0 infections is unlikely to be real world relevant (see discussion below) and doesn't leverage the UK specific model. If the authors chose to pursue this it is better to spin this off into another paper that includes theoretical analysis

similar to (<https://personalpages.manchester.ac.uk/staff/thomas.house/blog/why-zero-is-so-hard.html>)

(4) The spatial heterogeneity claims need to be strengthened with statistical analysis

Major comments about the model itself:

1: The authors need a very clear table of all the parameters of the model and whether these are fixed or estimated (an extended Table 3) and this should come much earlier in the paper.

2: I don't understand why the contact matrix from Mossong et al was binned into coarser age groups than the 5-year groups in original data. This seems entirely unnecessary for the purpose of this model yet incurs the cost of considerably under-estimating heterogeneity and thus overestimating epidemic size. I am in particular concerned about the binning of 20-60 year olds into one group. The number of contacts vary by several fold within that age range. The authors should re-run their analysis with the original bins. I expect that many of the key results regarding mitigation will be strengthened.

3: Much of the results and the discussion is devoted to asymptomatic infections. The authors state at the beginning: "The probability p that an infected individual develops symptoms is an important parameter for epidemic dynamics." It is not clear to me why this is universally the case without some qualifying assumption. These assumptions are never stated.

The importance of asymptomatic cases has to be attributed to:

- Asymptomatic individuals are less infectious
- Asymptomatic individuals recover more quickly
- Asymptomatic infections are less detectable.

If none of these hold then modeling asymptomatic infections is irrelevant. The authors should qualify which of these are being modeled, and where the alternative parameters come from. The overall infectiousness and recovery rates can be easily sourced from literature but how were the alternative asymptomatic parameters determined. These should be added with citations to the master parameter table.

Comments regarding the analysis

1: Overall I think that analyses claiming to produce accurate estimates of real world parameters need to show parameter/model robustness and also predictive value. At the very least, the authors should acknowledge that absolute numbers of infections or deaths computed by the model are not statements of fact. While I think it is reasonable to compare mitigation strategies within the context of this model saying that X number of people will die under scenario Y is not appropriate. These numbers are only as accurate as the underlying assumptions about model parameters which are uncertain. Furthermore there are many sources of unknown heterogeneity and this will in general reduce the real world epidemic size over what is predicted. The authors do not make an attempt to fit unmodeled heterogeneity (as is done in <https://www.medrxiv.org/content/10.1101/2020.04.27.20081893v3>) and I don't think this is necessary here but it does imply that statements about epidemic size and number of deaths cannot be taken as facts about the real world. Overall, I find this part of the analysis the least compelling and it distracts with the much more interesting (IMO) analysis of mitigation strategies at the end.

2: A related or separate point is that the following statement as it is misleading and undermines the otherwise high standard of scientific rigor in this paper.

"We estimate that, in absence of social distancing and confinement measures, the number of fatalities in England may have exceeded 216,000 by August 1, 2020, indicating that the lockdown has saved more than 174,000 lives." Besides the more global point (1) above it is not possible to say that X number of lives have been saved since this depends on the future number of deaths. Counterfactual claims are generally very difficult to support and should in my opinion be avoided altogether.

3: There are multiple issues with the figures throughout the paper.

-As a general rule the legend of the figure should be completely self sufficient in order to understand the content (What is A,H, and F in Figure 19?) The reader should not have to look at the text. Ideally, the authors would also give one or two sentences describing the point of the figure. For many figures in the paper I don't know what the reader should take away from it.

-Many of the figures are missing legend labels (perhaps a technical error with PDF generation???)

-There are too many figures. I suggest focusing on at most 7 highly compelling figures and making the rest supplemental.

-Some specific comments are below under "minor comments" but as there were a lot of omissions in the figures the list is likely not exhaustive and I suggest checking each one carefully.

4: The analysis presented in Figure 2 need to be fleshed out more. As far as I can tell the method does not return a posterior distribution. Can the authors determine if these values are indeed statistically different. It would be nice to back up these results with some external data. One of the regions with the highest multiplier (small region NNW of London) doesn't seem to correspond to any known population center, how should we interpret this result?

5: The analysis presented in Figure 5 and Table 4 is not convincing. The authors purport to show some variation in the regional multiplier. Even if it could be statistically shown that these are indeed different (see comment about Figure 2) the interpretation is not straight forward. Certainly, it is premature to call this "compliance" as it is not clear if these values are determined by relative changes in behavior rather than baseline variation in behavior/social norms. For example, for communities with relatively small rates of recreational socializing the multiplicative effect of mitigations will also be smaller since there is some minimal amount of necessary socializing that must persist. Overall I found this section of the paper to be among the least compelling.

6: The discussion of how it is possible to have a long run of 0 cases while the epidemic is progressing (section 4.2) should be shortened or removed. It is obvious that this *can* happen for some parameters (such as very low detection rate) but it also doesn't appear to be relevant to the real world. The authors bring up S. Korea as an example but cases never went to 0 there. I am not aware of a location where community acquired cases went to 0 and then re-emerged from a community source. While there was some speculation about the New Zealand second wave genomic analysis strongly favors the re-introduction hypothesis.

7. Authors need to provide source code, input data and examples.

Minor comments

-In equation 4.2 the authors use what they refer to as \bar{i} -- The (average) infection fatality rate. However, this is strongly dependent on the age distribution of the infected individuals. How was this taken into account?

-The analysis in Figure 9 seems to be considerably influenced by the 7 day periodicity (in the correlation is locally maximal when the periodicity is aligned, giving local peaks at 1, 7, and 14). It would be better to correlate the window average.

-Figure 3a is not explained at all.

-In figure 3b the simulation and the data should span the same interval on the x-axis.

-What is the pink region in Figure 8

-What is on the X-axis in Figure 10.

-Figure 16 is not referenced in the text. It is unclear what it shows. If these are data fits it would be better to show a per-region scatter plot against real data. The map display, while visually pleasing is not informative.

-I don't understand the purpose of Figure 17. Is there supposed to be a difference between (a) and (b) -- it is not visible by eye. Also legend labels are missing.

-I don't understand figure 18. There are no labels on the legend. I am assuming the orange line is the fit. Why does the first panel show results of multiple simulations but only an average is shown for the second panel. Why does the simulation not go for as long as the data?

-I don't see any purpose to figures 19-22 other than to show that the models are different. There is no systematic differences and the results are not compared to real data.

-small thing but in 2.6 I would change "man×day" to "person×day" since that most accurately reflects the correct unit.

-The authors assume that adaptive policies monitor case numbers. (As an aside I would avoid using R to denote this threshold to avoid confusion with the reproductive number). However, it is also possible to monitor hospitalizations that are less subject to reporting probability. This should be discussed.

Reviewer: 2

Comments to the Author(s)

The paper presents an extended compartmental model for SARS-COV-2 in the UK stratified by both age and region. The model is then used to assess the effectiveness of different control measures in reducing disease impact. This metric is then considered against the social cost of each set of measures in order to determine a sense of efficiency.

I generally found the article to be well presented and insightful with sensible modelling assumptions throughout. I believe the assessment of control measures will be of interest to others and have just a few suggestions for improvement prior to publication.

The authors spend a long time in the early parts of the paper (sections 3-6) discussing model fitting and presenting the benefits of a metapopulation approach. While much of this is certainly useful and informative, such models are certainly not new and similar approaches have already been extensively applied to the Covid pandemic. What this paper adds to the discussion is in its

assessment of control measures and I wonder whether it would be beneficial to move a lot of this early discussion to appendices or a supplement. The paper is not short and I think this would help give it a stronger message.

As far as the model itself is concerned, I thought the greatest deficiency for its use in the context of epidemic control, was the lack of any quarantining dynamic. Having symptomatic individuals (or even exposed households) reducing their contact may have a huge impact. Though I recognise no model is perfect, it may be worth acknowledging this for future inclusion.

In section 6.1 I was a little unclear on the length of the period of social distancing after lockdown. I assume no matter the level of compliance, if this period is long enough the social cost would be greater than that of the lockdown itself? Could this be clarified?

In section 7.1 there is a discussion of increasing the testing capacity. Is the effect of this identical to changing the triggering thresholds? Since the testing capacity increases over time then might there be some temporal effects on the triggering thresholds? This may be worth mentioning.

Finally a different colouring (over a larger spectrum) would be beneficial for the map figures, in their current form they appear rather homogenous.

===PREPARING YOUR MANUSCRIPT===

===PREPARING YOUR REVISION IN SCHOLARONE===

Author's Response to Decision Letter for (RSOS-201535.R0)

See Appendix A.

RSOS-201535.R1 (Revision)

Review form: Reviewer 2

Is the manuscript scientifically sound in its present form?

Yes

Are the interpretations and conclusions justified by the results?

Yes

Is the language acceptable?

Yes

Do you have any ethical concerns with this paper?

No

Have you any concerns about statistical analyses in this paper?

No

Recommendation?

Accept with minor revision (please list in comments)

Comments to the Author(s)

I thank the authors for taking the time to address my comments, and find the revised manuscript to be greatly improved.

Their response however has highlighted to me one additional issue. It is widely understood that an asymptomatic individual will spread the virus to a greatly reduced degree compared to a symptomatic case. I had assumed this was incorporated into the model via the parameter κ in equation 2.2. Following clarification on this point however, it seems that this parameter is in fact used to reduce spread from symptomatic individuals via a quarantine dynamic.

The issue with neglecting to include reduction in transmission from asymptomatics (which may be as great as a factor of 10) is it results in a far higher weighting in the importance of spread amongst younger age groups with high degrees of contact.

Due to the considerable amount of uncertainty we still have regarding all disease parameters for Covid, I don't believe this issue entirely invalidates the results, but I do think it significant enough that it should be highlighted in the text as a limitation of the model.

Decision letter (RSOS-201535.R1)

Dear Professor Cont,

On behalf of the Editors, we are pleased to inform you that your Manuscript RSOS-201535.R1 "Modelling COVID-19 contagion: risk assessment and targeted mitigation policies" has been accepted for publication in Royal Society Open Science subject to minor revision in accordance with the referees' reports. Please find the referees' comments along with any feedback from the Editors below my signature.

Please submit your revised manuscript and required files (see below) no later than 7 days from today's (ie 02-Mar-2021) date. Note: the ScholarOne system will 'lock' if submission of the revision is attempted 7 or more days after the deadline. If you do not think you will be able to meet this deadline please contact the editorial office immediately.

on behalf of Professor Tim Rogers (Associate Editor) and Mark Chaplain (Subject Editor)
openscience@royalsociety.org

Subject Editor Comments to Author (Professor Mark Chaplain):

Comments to the Author:

In addition to the comments of the reviewer and Associate Editor, please remove the "Contents" page in the final revised version - thank you.

Associate Editor Comments to Author (Professor Tim Rogers):
Comments to the Author:

If it is possible/practical it would be good to re-run some of the analysis with a modified infection rate for asymptomatic spreaders to check robustness to this parameter. Otherwise, a suitable modification to the text to highlight this important caveat will be needed.

Reviewer comments to Author:

Reviewer: 2

Comments to the Author(s)

I thank the authors for taking the time to address my comments, and find the revised manuscript to be greatly improved.

Their response however has highlighted to me one additional issue. It is widely understood that an asymptomatic individual will spread the virus to a greatly reduced degree compared to a symptomatic case. I had assumed this was incorporated into the model via the parameter κ in equation 2.2. Following clarification on this point however, it seems that this parameter is in fact used to reduce spread from symptomatic individuals via a quarantine dynamic.

The issue with neglecting to include reduction in transmission from asymptomatics (which may be as great as a factor of 10) is it results in a far higher weighting in the importance of spread amongst younger age groups with high degrees of contact.

Due to the considerable amount of uncertainty we still have regarding all disease parameters for Covid, I don't believe this issue entirely invalidates the results, but I do think it significant enough that it should be highlighted in the text as a limitation of the model.

===PREPARING YOUR MANUSCRIPT===

===PREPARING YOUR REVISION IN SCHOLARONE===

Author's Response to Decision Letter for (RSOS-201535.R1)

See Appendix B.

Decision letter (RSOS-201535.R2)

Dear Professor Cont,

It is a pleasure to accept your manuscript entitled "Modelling COVID-19 contagion: risk assessment and targeted mitigation policies" in its current form for publication in Royal Society Open Science.

COVID-19 rapid publication process:

We are taking steps to expedite the publication of research relevant to the pandemic. If you wish, you can opt to have your paper published as soon as it is ready, rather than waiting for it to be published the scheduled Wednesday.

This means your paper will not be included in the weekly media round-up which the Society sends to journalists ahead of publication. However, it will still appear in the COVID-19 Publishing Collection which journalists will be directed to each week (<https://royalsocietypublishing.org/topic/special-collections/novel-coronavirus-outbreak>).

If you wish to have your paper considered for immediate publication, or to discuss further, please notify openscience_proofs@royalsociety.org and press@royalsociety.org when you respond to this email.

Please see the Royal Society Publishing guidance on how you may share your accepted author manuscript at <https://royalsociety.org/journals/ethics-policies/media-embargo/>. After publication, some additional ways to effectively promote your article can also be found here

<https://royalsociety.org/blog/2020/07/promoting-your-latest-paper-and-tracking-your-results/>.

on behalf of Professor Tim Rogers (Associate Editor) and Mark Chaplain (Subject Editor)
openscience@royalsociety.org

RSOS-201535: Modelling COVID-19 contagion: Risk assessment and targeted mitigation policies

Rama CONT

Artur KOTLICKI

Renyuan XU

November 2020

We thank both reviewers for their detailed and constructive comments and insights, which we have carefully taken into account when revising our manuscript.

In summary, we have

- re-estimated the model with 16 age groups instead of only 4 age groups and redone all the analysis with this more granular model, as suggested by Reviewer 1;
- removed 15 figures from the paper and shortened in by 10 pages;
- removed references to absolute number of fatalities in Section 1;
- removed claims related to variation of compliance levels for social distancing across regions;
- updated our estimate of reporting probability in Section 4 using more recent data, which confirms the increase in reporting probabilities following an increase in testing across the UK;
- deleted the section on 'Counterfactual analysis: no intervention' (previously Section 5);
- corrected several typos pointed out by reviewers.

We hope the revised –and much improved– version will be deemed suitable for publication in OPEN SCIENCE.

We have also made available an interactive online app implementing the model described in the paper, which may be interesting as an online add-on for the reviewers and readers for further exploring the model:

<http://covid19.kotlicki.pl>

The following sections contain detailed responses to comments by the reviewers. Reviewers remarks are in italics, while our responses are in straight font.

1 Response to Reviewer 1

This paper presents a SIR-like model of COVID dynamics in the UK that incorporates social structure and geographic heterogeneity. Overall, the model is well put together and the consideration the authors give to spatial heterogeneity is a valuable contribution. The authors use this model to make a medley of different points including: (1) estimates of real world values (such as the total number of people infected),

(2) counterfactual estimates (how many lives the lock-down saved), (3) theoretical consideration (how many days it is possible to observe 0 cases while the epidemic is ongoing) (4) heterogeneity in model parameters across regions and (5) effects of different intervention strategies. Of these 5 points by (5) is by far the most compelling. The authors do a thorough analysis of mitigation strategies and also quantify the social cost which has seen little quantitative discussion in related work. [...] Points 1-4 should be either shortened/removed or considerably strengthened and/or turned into a separate manuscripts.

We thank the reviewer for the careful reading of the manuscript and the detailed feedback. We have considerably shortened points (1), (2) and (4) as suggested to better emphasise (5). We have left (3) as it responds to other points raised by the reviewer (see below). Overall this has shortened the paper by 10 pages.

With regards to specific analyses (1-4) my criticism in brief is:

1. *Estimates of real world values should be subjected to extensive robustness analysis and predictive (forecasting) validation. I believe this is beyond the scope of this paper.*

We agree that robustness check on *all* parameters is not feasible in a single paper, but the paper does contain predictive validation, region by region, using March 15 -May 31 as estimation period and June 1-Aug 1 as out-of-sample validation period. Robustness analysis is performed on several (though not all) parameters, including the asymptomatic ratio and the level of granularity (see last section of this document).

2. *Counterfactual estimates are generally difficult to support. Since the epidemic is not over we cannot purport to know how many lives were saved by the lockdown since it depends the future epidemic trajectory.*

We agree and we have toned down this aspect, and deleted what was previously Section 5 (Counterfactual analysis) also following your other remarks and those of Reviewer 2. We have now emphasised comparative analysis of policies rather than absolute forecasts.

3. *The discussion about the possible number of days with 0 infections is unlikely to be real world relevant (see discussion below) and doesn't leverage the UK specific model. If the authors chose to pursue this it is better to spin this off into another paper that includes theoretical analysis similar to (<https://personalpages.manchester.ac.uk/staff/thomas.house/blog/why-zero-is-so-hard.html>)*

A key aspect of the feedback policies is the fact that the state variables are not directly observed but 'sampled' with some reporting probability $\pi < 1$. The issue here is one of control with partial observation, whereas the analysis alluded to by the reviewer is related to the actual extinction probability of the infection, which is not what we discuss here.

4. *The spatial heterogeneity claims need to be strengthened with statistical analysis.*

Our goal is not to demonstrate specific patterns in spatial heterogeneity; we have modified the text to remove any assertion which may be interpreted as a 'significant difference' in behavior or compliance across regions.

Major comments about the model itself

1. *The authors need a very clear table of all the parameters of the model and whether these are fixed or estimated (an extended Table 3) and this should come much earlier in the paper.*

Point taken. We have grouped all model parameters in Tables 4 and 5, indicating the sources (if any) or the Appendix where estimates are given.

2. *I don't understand why the contact matrix from Mossong et al was binned into coarser age groups than the 5-year groups in original data. This seems entirely unnecessary for the purpose of this model yet incurs the cost of considerably under-estimating heterogeneity and thus overestimating epidemic size. I am in particular concerned about the binning of 20-60 year olds into one group. The number of contacts vary by several fold within that age range. The authors should re-run their analysis with the original bins. I expect that many of the key results regarding mitigation will be strengthened.*

Following this remark we have re-done all the analysis with the original (16) 5-year age groups used by Mossong et al. Interestingly, the results are quite similar to our original model with 4 age groups. However, some quantitative differences may emerge when assessing the impact of targeted policies. Figure 1 show an example of such differences. (Details of the models compared are given in Section 3 below). The dynamics of infections, cases and fatalities are rather insensitive to the demographic granularity but some differences in the impact of targeted policies are visible.

In the revised version, we have followed the recommendation of the reviewer and retained the more granular model with 16 age groups.

Figure 1: Impact of demographic granularity on model outcomes: example of centralised adaptive policy with $R_{on} = 4$, $R_{off} = 0.4 \times R_{on}$, $m = 0.5$ and no shielding.

3. *Much of the results and the discussion is devoted to asymptomatic infections. The authors state at the beginning: "The probability p that an infected individual develops symptoms is an important*

parameter for epidemic dynamics.” It is not clear to me why this is universally the case without some qualifying assumption. These assumptions are never stated.

The importance of asymptomatic cases has to be attributed to:

- Asymptomatic individuals are less infectious
- Asymptomatic individuals recover more quickly
- Asymptomatic infections are less detectable.

If none of these hold than modeling asymptomatic infections is irrelevant. The authors should qualify which of these are being modeled, and where the alternative parameters come from. The overall infectiousness and recovery rates can be easily sources from literature but how were the alternative asymptomatic parameters determined. These should be added with citations to the master parameter table.

In our model, asymptomatic infections are not detectable so asymptomatic individuals do not self-isolate and maintain the same level of social contact as non-infected ones. This effect is represented by the coefficient $\kappa < 1$ in the force of infection, which represents a reduction of contacts for symptomatic individuals, an effect not present for asymptomatic individuals. This clearly affects epidemic dynamics. Independently of this effect, the presence of asymptomatic individuals affects compartmental dynamics as the sum over compartments is constant.

The source for the asymptomatic ratios is the ONS study [5] which shows that a high proportion –possibly more than 70%– of COVID-19 carriers may be asymptomatic, so clearly this is a relevant feature to be included in the model. This study is based on a larger sample than previous estimates such as the Diamond Princess study, and the estimates are adjusted for the UK population distribution, which is not the case for previous studies.

There is a wide range for estimated of symptomatic ratios (various estimates were/are discussed in Sec. 3.3) and we have carried out a corresponding robustness analysis systematically, computing all projections under ‘high’ and ‘low’ assumptions for the asymptomatic ratios. As shown in Section 5 and 6, the results do depend on the asymptomatic ratio, showing that this is an important parameter.

Comments regarding the analysis

1. Overall I think that analyses claiming to produce accurate estimates of real world parameters need to show parameter/model robustness and also predictive value. At the very least, the authors should acknowledged that absolute numbers of infections or deaths computed by the model are not statements of fact. While I think it is reasonable to compare mitigation strategies within the context of this model saying that X number of people will die under scenario Y is not appropriate. These numbers are only as accurate as the underlying assumptions about model parameters which are uncertain. Furthermore there are many sources of unknown heterogeneity and this will in general reduce the real world epidemic size over what is predicted. The authors do not make an attempt to fit unmodeled heterogeneity (as is done in <https://www.medrxiv.org/content/10.1101/2020.04.27.20081893v3>) and I don't think this is necessary here but it does imply that statements about epidemic size and number of deaths cannot be taken as facts about the real world. Overall, I find this part of the analysis the least compelling and it distracts with the much more interesting (IMO) analysis of mitigation strategies at the end.

We acknowledge that the main output of the model are not the absolute projections for fatalities or infections but the comparative analysis of the models. We have removed such statements from the ‘Main findings’ section. However, a quantitative comparison cannot be done without stating

at some point numerical outcomes for various policies, and this is the sole purpose of the numerical values reported in various tables in the paper.

2. *A related or separate point is that the following statement as it is misleading and undermines the otherwise high standard of scientific rigor in this paper. "We estimate that, in absence of social distancing and confinement measures, the number of fatalities in England may have exceeded 216,000 by August 1, 2020, indicating that the lockdown has saved more than 174,000 lives." Besides the more global point (1) above it is not possible to say that X number of lives have been saved since this depends on the future number of deaths. Counterfactual claims are generally very difficult to support and should in my opinion be avoided altogether.*

We concede this point and have removed this statement from Section 1 and deleted the section on 'counterfactual analysis' in its entirety.

3. *There are multiple issues with the figures throughout the paper. -As a general rule the legend of the figure should be completely self sufficient in order to understand the content (What is A,H, and F in Figure 19?) The reader should not have to look at the text. Ideally, the authors would also give one or two sentences describing the point of the figure. For many figures in the paper I don't know what the reader should take away from it.*

We apologise for the lack of clarity in the figures. We have added legends and detailed captions to all figures.

-Many of the figures are missing legend labels (perhaps a technical error with PDF generation???)

Legend labels have now been added to all figures.

-There are too many figures. I suggest focusing on at most 7 highly compelling figures and making the rest supplemental.

We have deleted 14 figures, reducing the total number of figures to 30.

4. *The analysis presented in Figure 2 need to be fleshed out more. As far as I can tell the method does not return a posterior distribution. Can the authors determine if these values are indeed statistically different. It would be nice to back up these results with some external data. One of the regions with the highest multiplier (small region NNW of London) doesn't seem to correspond to any known population center, how should we interpret this result?*

d_r is estimated by indirect inference [3], which does yield confidence intervals but we have not reported them here as it does not serve any purpose for our analysis.

We have added Table 2, which lists the regions with highest regional adjustment factors, as well as some of their characteristics (inward/outward mobility, population density). The regions with highest multiplier are South Teesside, Croydon and Solihull. As shown in Table 2, regions with high d_r may correspond to regions with high commuting rates or high population density. But it is not the purpose of our study to focus on regional characteristics, which would require other types of socio-economic data, and we do not claim to explain these differences.

5. *The analysis presented in Figure 5 and Table 4 is not convincing. The authors purport to show some variation in the regional multiplier. Even if it could be statistically shown that these are indeed different (see comment about Figure 2) the interpretation is not straight forward. Certainly, it is premature to call this "compliance" as it is not clear if these values are determined by relative changes in behavior rather than baseline variation in behavior/social norms. For example, for communities with relatively small rates of recreational socializing the multiplicative effect of mitigations will also be smaller since there is some minimal amount of necessary socializing that must persist. Overall I found this section of the paper to be among the least compelling.*

We have removed Figure 5 as well as any reference to ‘compliance’ as suggested by the reviewer. We do not claim the regional differences in the l_r values to be significant. In fact, they are all quite close to 10% (representing a 90% drop in social contacts during lockdown).

6. *The discussion of how it is possible to have a long run of 0 cases while the epidemic is progressing (section 4.2) should be shortened or removed. It is obvious that this *can* happen for some parameters (such as very low detection rate) but it also doesn't appear to be relevant to the real world. The authors bring up S. Korea as an example but cases never went to 0 there. I am not aware of a location where community acquired cases went to 0 and then re-emerged from a community source. While there was some speculation about the New Zealand second wave genomic analysis strongly favors the re-introduction hypothesis.*

This section has been shortened and one figure removed.

The parameters used in the examples are not extreme or unrealistic parameters: they are the ones estimated from the data, so we do not think this example is far-fetched. It results from a combination of

- partial observability of the total number of cases (in absence of widespread testing), and
- stochastic dynamics of the model, which lead to the possibility of random flare-ups.

Note that we are not referring to the situation where cases go *zero* but to the situation where cases go *undetected*: even with a detection probability of 20%, which is quite high, the probability of observing a second peak after 60 days without reported cases is found to be 60%. This is not a small probability and corresponds to the fact that random flare-ups may originate from a small group of undetected cases.

7. *Authors need to provide source code, input data and examples.*

The source code and data have been now deposited on a public repository on GitHub:

<https://github.com/RenyuanXu/COVID-19>

Moreover we have implemented an easy-to-use online simulation app for running scenario simulation and comparative analysis of mitigation policies: <http://covid19.kotlicki.pl> which readers can use to explore other scenarios/policies than those presented in the paper.

Minor comments

In equation 4.2 the authors use what they refer to as \bar{f} – The (average) infection fatality rate. However, this is strongly dependent on the age distribution of the infected individuals. How was this take into account?

This is a population-weighted average fatality rate averaged across age groups using England demographics. The same value was used in Ferguson et al. [2].

- The analysis in Figure 9 seems to be considerably influenced by the 7 day periodicity (in the correlation is locally maximal when the periodicity is aligned, giving local peaks at 1, 7 and 14). It would be better to correlate the window average.

Thanks for this remark. We have redone the averaging as suggested (see new Figure 7).

Figure 3a is not explained at all

We have deleted Fig 3a

In figure 3b the simulation and the data should span the same interval on the x-axis.

We have deleted Fig 3b

What is the pink region in Figure 8

We have deleted the pink region in Fig 8 (now Fig 6).

What is on the X-axis in Figure 10.

This is now Figure 7. This figure displays the estimated reporting ratio as a function of time, estimated using a rolling window.

Figure 16 is not referenced in the text. It is unclear what it show. If these are data fits it would be better to show a per-region scatter plot against real data. The map display, while visually pleasing is not informative.

Figure 16 has been deleted. In fact the entire section on counterfactual analysis was deleted.

I don't understand the purpose of Figure 17. Is there supposed to be a difference between (a) and (b) – it is not visible by eye. Also legend labels are missing.

Figure 17 has been deleted. In fact the entire section on counterfactual analysis was deleted.

I don't understand figure 18. There are no labels on the legend. I am assuming the orange line is the fit. Why does the first panel show results of multiple simulations but only an average is shown for the second panel. Why does the simulation not go for as long as the data?

Figure 18 has been deleted. In fact the entire section on counterfactual analysis was deleted.

I don't see any purpose to figures 19-22 other than to show that the models are different. There is no systematic differences and the results are not compared to real data.

Figures 19-22 has been deleted as part of the section on counterfactual analysis.

-small thing but in 2.6 I would change "man×day" to "person×day" since that most accurately reflects the correct unit.

Many thanks: we have made this change.

-The authors assume that adaptive policies monitor case numbers. (As an aside I would avoid using R to denote this threshold to avoid confusion with the reproductive number). However, it is also possible to monitor hospitalizations that are less subject to reporting probability. This should be discussed.

We have modified the notation of thresholds to $B_{\text{on}}, B_{\text{off}}$. Case numbers are in fact widely reported and used by public health authorities to monitor the epidemic. Data on hospitalizations is not available to us. We note that, if one assumes that a fraction π of infected are hospitalized, then we end up with a similar censoring problem as with reported versus total cases.

2 Response to Reviewer 2

I generally found the article to be well presented and insightful with sensible modelling assumptions throughout. I believe the assessment of control measures will be of interest to others and have just a few suggestions for improvement prior to publication. The authors spend a long time in the early parts of the paper (sections 3-6) discussing model fitting and presenting the benefits of a metapopulation approach. While much of this is certainly useful and informative, such models are certainly not new and similar approaches have already been extensively applied to the Covid pandemic. What this paper adds to the discussion is in its assessment of control measures and I wonder whether it would be beneficial to move a lot of this early discussion to appendices or a supplement. The paper is not short and I think this would help give it a stronger message.

We thank the reviewer for these detailed remarks. We have shortened the early parts and, in particular, deleted the section on 'counterfactual scenario: no intervention' (previously Sec 5), to focus on mitigation policies as suggested. This has shortened the paper by 10 pages.

As far as the model itself is concerned, I thought the greatest deficiency for its use in the context of epidemic control, was the lack of any quarantining dynamic. Having symptomatic individuals (or even exposed households) reducing their contact may have a huge impact. Though I recognise no model is perfect, it may be worth acknowledging this for future inclusion.

In fact we do have a quarantine effect: this is the role of the coefficient $\kappa < 1$ in the force of infection Eq (2.2). We assume that social contacts for infected individuals are lower than for others by a factor $\kappa < 1$; this may be interpreted as quarantine i.e. avoidance of social contact by a fraction κ of infectious individuals. In the simulations we use $\kappa = 0.5$

In section 6.1 I was a little unclear on the length of the period of social distancing after lockdown. I assume no matter the level of compliance, if this period is long enough the social cost would be greater than that of the lockdown itself? Could this be clarified?

We have assumed social distancing after/between lockdowns. The level of reduction in social contacts is parameterized by the coefficient $0 \leq m \leq 1$, with $m = 0$ corresponding to lockdown (estimated $\sim 90\%$ reduction in contact rates) and $m = 1$ to 'normal' contacts. The social contact rates are then scaled by m . The reviewer is correct to state that a long period of social distancing can entail a social cost (i.e. reduction on social contacts) greater than a short lockdown. That is precisely why an efficiency analysis in terms of social cost vs health outcome is useful (and its outcome far from obvious).

In section 7.1 there is a discussion of increasing the testing capacity. Is the effect of this identical to changing the triggering thresholds? Since the testing capacity increases over time then might there be some temporal effects on the triggering thresholds? This may be worth mentioning.

The two issues are related but not identical. If one increases the test capacity, say by a factor 2, then this will double the number of reported cases so the triggering threshold should be also doubled in order to retain a similar policy. However increasing the testing capacity also reduces the fraction of non-detected cases so makes the policy more effective.

Finally a different colouring (over a larger spectrum) would be beneficial for the map figures, in their current form they appear rather homogeneous.

We have used a broader range of colours, going from yellow to red to black.

3 Impact of demographic granularity

In response to the comment by Reviewer 1, we have re-calibrated our model (previously using 4 age groups) with a more detailed model with the 16 age groups used by Mossong et al. [4]). We have also looked at other variations, e.g. splitting the age group 20-59 into 2 subgroups, etc.

The results are quite similar to our original model with 4 age groups. However, some quantitative differences may emerge when assessing the impact of targeted policies. Figure 1 shows an example of such differences. The dynamics of infections, cases and fatalities are rather insensitive to the demographic granularity but some differences in the impact of targeted policies are visible.

In the revised version, we have followed the recommendation of the reviewers and retained the more granular model with 16 age groups.

This section provides details on the different levels of granularity used, whose results are compared in Figure 1 (now added in the paper in Section 6.1).

3.1 Four age groups

We consider the following four age groups $[0, 20)$, $[20, 60)$, $[60, 70)$ and $70+$.

Age group	[0,20)	[20,60)	[60,70)	[70, 100)
Size	13.21 M 23.6%	29.43 M 52.6%	5.86 M 10.5%	7.40 M 13.3%

Table 1: Age group distribution for England, 2019.

We assume people with age between 20 and 60 will travel for work (age groups 2), hence the the force of infection $\lambda_t(r, a)$ is given by

$$\lambda_t(r, a) = \alpha \sum_{a' \neq 2} \sigma_{a,a'}^r(t) \frac{\kappa I_t(r, a') + A_t(r, a')}{N(r, a')} + \alpha \sigma_{a,2}^r(t) \sum_{r'=1}^K M_{r,r'}(t) \frac{\kappa I_t(r', 2) + A_t(r', 2)}{N(r', 2)}, \quad (1)$$

and the contact matrix is given by

$$\sigma = \begin{pmatrix} 7.88 & 5.3 & 0.44 & 0.37 \\ 2.40 & 7.52 & 0.84 & 0.58 \\ 1.12 & 4.72 & 1.17 & 1 \\ 0.80 & 2.75 & 0.83 & 1.34 \end{pmatrix}. \quad (2)$$

Figure 2: Baseline social contact matrices with 4 age groups.

Age group	0-19	20-59	60-69	≥ 70
	p_1	p_2	p_3	p_4
Low estimate	0.055	0.25	0.375	0.375
High estimate	0.11	0.5	0.75	0.75

Table 2: Age-dependent symptomatic ratios. Source: the Office for National Statistics [5].

Age group	[0,20)	[20,40)	[40,60)	[60,70)	[70, 100)
(%)	0.01	0.15	0.45	2.1	6.5

Table 3: Fatality rate.

3.2 Five age groups

We divide age group [20, 60) into two groups [20, 40) and [40, 60).

Age group	[0,20)	[20,40)	[40,60)	[60,70)	[70, 100)
Size	13.21 M 23.6%	14.8 M 26.4	14.7 M 26.2	5.86 M 10.5%	7.40 M 13.3%

Table 4: Age group distribution for England, 2019.

We assume people with age between 20 and 60 will travel for work (age groups 2-3), hence the force of infection $\lambda_t(r, a)$, which measures the rate of exposure at location r for age group a , is given by

$$\begin{aligned}
\lambda_t(r, a) = & \alpha \sum_{a' \neq 2} \sigma_{a,a'}^r(t) \frac{\kappa I_t(r, a') + A_t(r, a')}{N(r, a')} \\
& + \alpha \sigma_{a,2}^r(t) \sum_{r'=1}^K M_{r,r'}(t) \frac{\kappa I_t(r', 2) + A_t(r', 2)}{N(r', 2)} \\
& + \alpha \sigma_{a,3}^r(t) \sum_{r'=1}^K M_{r,r'}(t) \frac{\kappa I_t(r', 3) + A_t(r', 3)}{N(r', 3)},
\end{aligned} \tag{3}$$

and the contact matrix is given by

$$\sigma = \begin{pmatrix} 7.88 & 3.12 & 2.17 & 0.44 & 0.37 \\ 2.79 & 4.85 & 3.15 & 0.78 & 0.36 \\ 1.98 & 3.22 & 3.78 & 0.90 & 0.81 \\ 1.11 & 2.21 & 2.51 & 1.17 & 0.99 \\ 0.80 & 0.87 & 1.89 & 0.83 & 1.35 \end{pmatrix}. \tag{4}$$

Figure 3: Baseline social contact matrices with 16 age groups.

Age group	[0,20)	[20,40)	[40,60)	[60,70)	[70, 100)
(%)	0.01	0.25	0.45	2	6.5

Table 5: Fatality rate.

Age group	0-19	20-39	20-39	60-69	≥ 70
	p_1	p_2	p_3	p_4	p_5
Low estimate	0.055	0.2	0.25	0.375	0.375
High estimate	0.11	0.4	0.5	0.75	0.75

Table 6: Age-dependent symptomatic ratios. Source: the Office for National Statistics [5].

3.3 16 age groups

We consider 16 age groups with 5 years difference: $[0, 5)$, $[5, 10)$, $[10, 15)$, $[15, 20)$, $[20, 25)$, $[25, 30)$, $[30, 35)$, $[35, 40)$, $[40, 45)$, $[45, 50)$, $[50, 55)$, $[55, 60)$, $[60, 65)$, $[65, 70)$, $[70, 75)$, and $75+$.

Age group	$[0,5)$	$[5,10)$	$[10,15)$	$[15,20)$	$[20, 25)$	$[25, 30)$	$[30, 35)$	$[35, 40)$
Size (millions)	3.3	3.5	3.3	3.1	3.5	3.8	3.8	3.7
Fraction	5.9%	6.3%	5.9%	5.5%	6.2%	6.8%	6.8%	6.6%
Age group	$[40,45)$	$[45,50)$	$[50,55)$	$[55,60)$	$[60, 65)$	$[65, 70)$	$[70, 75)$	$[75, 100)$
Size (millions)	3.4	3.8	3.9	3.6	3.1	2.8	2.8	4.7
Fraction	6.0%	6.7%	7.0%	6.5%	5.5%	5.0%	4.9%	8.4%

Table 7: Age group distribution for England, 2019. Source: Eurostat [1].

We assume people with age between 20 and 60 will travel for work (age groups 5-12), hence the the force of infection $\lambda_t(r, a)$ is given by

$$\lambda_t(r, a) = \alpha \sum_{a' \notin \mathcal{W}} \sigma_{a,a'}^r(t) \frac{\kappa I_t(r, a') + A_t(r, a')}{N(r, a')} + \alpha \sum_{a' \in \mathcal{W}} \sigma_{a,a'}^r(t) \sum_{r'=1}^K M_{r,r'}(t) \frac{\kappa I_t(r', a') + A_t(r', a')}{N(r', a')} \quad (5)$$

where $\mathcal{W} = \{5, 6, 7, 8, 9, 10, 11, 12\}$.

Figure 4: Baseline social contact matrices with 16 age groups.

Age group	[0,5)	[5,10)	[10,15)	[15,20)	[20, 25)	[25, 30)	[30, 35)	[35, 40)
(%)	0.002	0.002	0.01	0.01	0.05	0.05	0.1	0.1
Age group	[40,45)	[45,50)	[50,55)	[55,60)	[60, 65)	[65, 70)	[70, 75)	[75, 80)
(%)	0.2	0.2	0.6	0.6	2.00	2.00	4.0	7.5

Table 8: Fatality rate.

Age group	[0,5)	[5,10)	[10,15)	[15,20)	[20, 25)	[25, 30)	[30, 35)	[35, 40)
Low estimate	0.075	0.075	0.05	0.05	0.15	0.15	0.21	0.21
High estimate	0.15	0.15	0.1	0.1	0.3	0.3	0.42	0.42
Age group	[40,45)	[45,50)	[50,55)	[55,60)	[60, 65)	[65, 70)	[70, 75)	[75, 80)
Low estimate	0.23	0.23	0.28	0.28	0.41	0.41	0.375	0.375
High estimate	0.45	0.45	0.56	0.56	0.82	0.82	0.75	0.75

Table 9: Age-dependent symptomatic ratios. Source: the Office for National Statistics [5].

References

- [1] Eurostat. Population on 1 January by age group, sex and NUTS 3 region [computer file]. European Commission. Downloaded from: https://appsso.eurostat.ec.europa.eu/nui/show.do?dataset=demo_r_pjangrp3&lang=en, January 2020. Table demo_r_pjangrp3.
- [2] Neil Ferguson, Daniel Laydon, Gemma Nedjati Gilani, Natsuko Imai, Kylie Ainslie, Marc Baguelin, Sangeeta Bhatia, Adhiratha Boonyasiri, Zulma Cucunuba-Perez, Gina Cuomo-Dannenburg, et al. Report 9: Impact of non-pharmaceutical interventions (npis) to reduce COVID-19 mortality and healthcare demand. 2020.
- [3] Christian Gourieroux, Alain Monfort, and Eric Renault. Indirect inference. *Journal of applied econometrics*, 8(S1):S85–S118, 1993.
- [4] Joel Mossong, Niel Hens, Mark Jit, Philippe Beutels, Kari Auranen, Rafael Mikolajczyk, Marco Massari, Stefania Salmaso, Gianpaolo Scalia Tomba, Jacco Wallinga, Janneke Heijne, Malgorzata Sadkowska-Todys, Magdalena Rosinska, and W. John Edmunds. Social contacts and mixing patterns relevant to the spread of infectious diseases. *PLOS Medicine*, 5(3):1–1, March 2008.
- [5] Office for National Statistics. Coronavirus (COVID-19) infections in the community in England: July 2020. Downloaded from: <https://www.ons.gov.uk/peoplepopulationandcommunity/healthandsocialcare/conditionsanddiseases/articles/coronaviruscovid19infectionsinthecommunityinengland/july2020>, January 2020.

Appendix B

RSOS-201535: Modelling COVID-19 contagion: Risk assessment and targeted mitigation policies

Rama CONT

Artur KOTLICKI

Renyuan XU

Second revision: March 2021

We are grateful to the Associate Editor and reviewer for their constructive comments regarding the differences in infection rates for symptomatic vs asymptomatic carriers.

We agree that this may be an important feature, so we have followed the suggestion of the reviewer and re-run the model with this feature, distinguishing

- the infection rate α_0 for asymptomatic carriers from
- the infection rate $\alpha_1 > \alpha_0$ for symptomatic carriers.

In terms of the model, introducing heterogeneous infection rates only affects the expression of the force of infection (Equation 2.2 in the paper). The rest is unaffected.

Estimation of symptomatic vs asymptomatic infection rates Previously we had estimated a single infection rate parameter (which was shown to be consistent with values used in most other papers on COVID modeling). Now this value is interpreted as an average value across symptomatic/asymptomatic carriers.

To estimate the heterogeneous infection rates, we have based our approach on the recent study [1] which estimates that, when adjusted for age and gender, the incidence of COVID-19 among close contacts of a symptomatic index case is 3.85 times higher than for close contacts of an asymptomatic carrier, which means

$$\alpha_1 \simeq 3.85\alpha_0$$

We explain in Sec 3.4 how we use this constraint together with previous estimates of the average infection rate to estimate α_0, α_1 in each age class. The results are shown in the table below.

Age group	[0,5)	[5,10)	[10,15)	[15,20)	[20, 25)	[25, 30)	[30, 35)	[35, 40)
α_0	0.045	0.045	0.048	0.048	0.039	0.039	0.034	0.034
α_1	0.174	0.174	0.185	0.185	0.148	0.148	0.132	0.132
Age group	[40,45)	[45,50)	[50,55)	[55,60)	[60, 65)	[65, 70)	[70, 75)	[75, 100)
α_0	0.033	0.033	0.031	0.031	0.025	0.025	0.027	0.027
α_1	0.128	0.128	0.118	0.118	0.098	0.098	0.102	0.102

Table 1: Age-dependent infection rates: symptomatic (α_1) vs asymptomatic (α_0).

Influence of different infection rates for symptomatic vs asymptomatic carriers Using the estimated values of infection rates $\alpha_0 < \alpha_1$, we have re-run scenario simulations and compared the results with the previous simulations with a homogeneous infection rate.

These differences are discussed in a new Section 5.3 **Impact of parameter uncertainty**.

Figures 18 and 19 illustrate the impact of having heterogeneous infection rates. Introducing different infection rates for symptomatic vs asymptomatic carriers reduces the estimated fatalities by 10-20% in the scenarios, but does not modify the conclusions regarding the comparison of different policies. For example in Section 5 (Figure 18), we still observe school closures to be less effective than restrictions on non-work/school gatherings ('no pubs').

We conclude from our new simulations that the policy comparisons, which are the main focus of the paper, are not affected by (although the numerical outcomes of the scenarios are of course affected by these and other parameters).

We hope the revised version will be deemed suitable for publication in OPEN SCIENCE.

References

- [1] Andrew A. Sayampanathan, Cheryl S Heng, Phua Hwee Pin, Junxiong Pang, Teoh Yee Leong, and Vernon J Lee. Infectivity of asymptomatic versus symptomatic COVID-19. *The Lancet*, 397(10269): 93–94, 2021.